# Assembly defects of human tRNA splicing endonuclease contribute to impaired pre-tRNA processing in pontocerebellar hypoplasia

Samoil Sekulovski [1,9], Pascal Devant [1,2,3,9], Silvia Panizza[4,9], Tasos Gogakos[5], Anda Pitiriciu[1], Katharina Heitmeier [1], Ewan Phillip Ramsay[6], Marie Barth [7], Carla Schmidt [7], Thomas Tuschl [5], Frank Baas[8], Stefan Weitzer[4], Javier Martinez [4✉] & Simon Trowitzsch [1✉]

Introns of human transfer RNA precursors (pre-tRNAs) are excised by the tRNA splicing endonuclease TSEN in complex with the RNA kinase CLP1. Mutations in TSEN/CLP1 occur in patients with pontocerebellar hypoplasia (PCH), however, their role in the disease is unclear. Here, we show that intron excision is catalyzed by tetrameric TSEN assembled from inactive heterodimers independently of CLP1. Splice site recognition involves the mature domain and the anticodon-intron base pair of pre-tRNAs. The 2.1-Å resolution X-ray crystal structure of a TSEN15–34 heterodimer and differential scanning fluorimetry analyses show that PCH mutations cause thermal destabilization. While endonuclease activity in recombinant mutant TSEN is unaltered, we observe assembly defects and reduced pre-tRNA cleavage activity resulting in an imbalanced pre-tRNA pool in PCH patient-derived fibroblasts. Our work defines the molecular principles of intron excision in humans and provides evidence that modulation of TSEN stability may contribute to PCH phenotypes.

---

[1] Institute of Biochemistry, Biocenter, Goethe University Frankfurt, Frankfurt/Main, Germany. [2] Ph.D. Program in Virology, Harvard Medical School, Boston, MA, USA. [3] Harvard Medical School and Division of Gastroenterology, Boston Children's Hospital, Boston, MA, USA. [4] Max Perutz Labs, Medical University of Vienna, Vienna Biocenter (VBC), Vienna, Austria. [5] Laboratory for RNA Molecular Biology, The Rockefeller University, New York, NY, USA. [6] The Institute of Cancer Research, London, United Kingdom. [7] Interdisciplinary research center HALOmem, Charles Tanford Protein Center, Institute of Biochemistry and Biotechnology, Martin Luther University Halle-Wittenberg, Halle, Germany. [8] Department of Clinical Genetics, Leiden University, Leiden, Netherlands. [9] These authors contributed equally: Samoil Sekulovski, Pascal Devant, Silvia Panizza. ✉email: javier.martinez@meduniwien.ac.at; trowitzsch@biochem.uni-frankfurt.de

All nuclear-encoded transfer RNAs (tRNAs) are processed and modified to create functional, aminoacylated tRNAs[1]. In humans, 28 out of 429 predicted high confidence tRNA genes contain introns that must be removed from precursor tRNAs (pre-tRNAs) by splicing[2,3] (http://gtrnadb.ucsc.edu/). Some iso-decoders, e.g. tRNA$^{Tyr}_{GTA}$, tRNA$^{Ile}_{TAT}$, and tRNA$^{Leu}_{CAA}$, are only encoded as intron-containing precursors, for which splicing is essential for their production[4]. Intron excision and ligation of the 5′ and 3′ tRNA exons is catalyzed by two multiprotein assemblies: the tRNA splicing endonuclease (TSEN)[5] and the tRNA ligase complex[6], respectively.

The human TSEN complex consists of two catalytic subunits, TSEN2 and TSEN34, and two structural subunits, TSEN15 and TSEN54, all expressed at very low copy numbers of ~100 molecules per cell[5,7]. TSEN2–54 and TSEN15–34 are inferred to form distinct heterodimers from yeast-two-hybrid experiments, however, a solution NMR structure of homodimeric TSEN15 has challenged the proposed model of TSEN assembly[8,9]. Based on their quaternary structure, archaeal tRNA endonucleases have been classified into four types, $\alpha_4$, $\alpha'_2$, $(\alpha\beta)_2$, and $\varepsilon_2$[10,11], whereas the eukaryotic tRNA endonucleases adopt a heterotetrameric αβγδ arrangement[5,8]. Homotetramer formation in archaeal $\alpha_4$-type endonucleases is mediated by a hydrophobic domain interface involving antiparallel β strands of two neighboring α subunits and interactions between a negatively charged L10 loop of one α subunit with a positively charged pocket of an opposing α subunit. These interactions are conserved in all four types of archaeal endonucleases and were also suggested to occur in eukaryotic endonucleases. In humans, TSEN2 and TSEN34 are each predicted to harbor a catalytic triad, composed of Tyr$^{369}$/His$^{377}$/Lys$^{416}$ in TSEN2 and Tyr$^{247}$/His$^{255}$/Lys$^{286}$ in TSEN34, responsible for cleavage at the 5′ and 3′ splice sites, respectively[5,8,12]. His$^{377}$ and His$^{255}$ are supposed to act as general acids at the scissile phosphates of the exon-intron junctions of pre-tRNAs[5,12,13]. Furthermore, TSEN54 was suggested to function as a molecular ruler measuring the distance from the mature domain of the tRNA to define the 5′ splice site[8,13–16].

The intron in pre-tRNAs is suggested to allow the formation of a double helix that extends the anticodon stem in the conventional tRNA cloverleaf structure and presents the 5′ and 3′ splice sites in single-stranded regions accessible to TSEN[17,18]. Such a bulge-helix-bulge (BHB) conformation was postulated to act as a universal recognition motif in archaeal pre-tRNA splicing allowing for intron recognition at various positions in pre-tRNAs[19]. In contrast, eukaryotic introns strictly locate one nucleotide 3′ to the anticodon triplet in the anticodon loop with varying sequence and length[3,14]. Experiments using yeast and *Xenopus* tRNA endonucleases showed that cleavage at the exon-intron boundaries requires the presence of an anticodon–intron (A–I) base pair that controls cleavage at the 3′ splice site besides positioning of the 5′ splice site via the mature domain of the pre-tRNA[14,20,21]. The X-ray crystal structure of an archaeal endonuclease with a BHB-substrate showed that two arginine residues at each active site form a cation-π sandwich with a flipped-out purine base of the pre-tRNA to position the substrate for an $S_N2$-type in line-attack[12,13,16]. However, biochemical experiments using the yeast endonuclease showed that the cation-π sandwich is only required for cleavage at the 5′ splice site, whereas it is dispensable for catalysis at the 3′ splice site[12].

Specific to mammals is the association of TSEN with the RNA kinase CLP1[5,22]. Mutations in CLP1 were shown to impair tRNA splicing in vitro and to cause neuropathologies involving the central and peripheral nervous system[23–25]. Mutations in all four subunits of the TSEN complex have been associated with the development of pontocerebellar hypoplasia (PCH), a heterogeneous group of inherited neurodegenerative disorders with prenatal onset characterized by cerebellar hypoplasia and microcephaly[26–31].

The most common mutation causing a type 2 PCH phenotype is a homozygous *TSEN54* c.919 G > T mutation that leads to an A$^{307}$S substitution in TSEN54[26,27,32]. Several other substitutions, e.g. S$^{93}$P in TSEN54, R$^{58}$W in TSEN34, Y$^{309}$C in TSEN2, and H$^{116}$Y in TSEN15, have also been identified as causative for PCH[26,30]. None of the described disease mutations is located in or in close proximity to the predicted catalytic sites of human TSEN, or in other highly conserved regions of the proteins, and how they contribute to disease development remains unclear. Here we present the biochemical and structural characterization of recombinant human TSEN. We analyze PCH-associated mutations at the structural and biochemical levels in reconstitution experiments and reveal biochemical features of the TSEN complex in PCH patient-derived cells.

## Results

**Assembly of recombinant human tRNA splicing endonuclease.** To gain functional insights into human TSEN/CLP1, we designed an expression vector series based on the MultiBac system[33] that allows combinatorial protein complex production in insect and mammalian cells by utilizing a CMV/p10 dual promoter[34] (Fig. 1a, b, and Supplementary Fig. 1a). Using this system, we were able to assemble and purify functional heterotetrameric TSEN and a heteropentameric complex including the RNA kinase CLP1 from infected insect cells (Fig. 1b, c and Supplementary Fig. 1b). The structural integrity of the purified complexes was verified by native mass spectrometry (MS), showing a stoichiometric TSEN2–15–34–54 heterotetramer (165.6 kDa) and a TSEN/CLP1 heteropentamer (213.0 kDa) (Fig. 1c, Supplementary Fig. 1b, and Supplementary Tables 1,2). These data are in line with a recent study showing reconstitution of TSEN/CLP1 from bacterial and eukaryotic expression hosts[35]. We also identified TSEN/CLP1 complexes harboring two CLP1 molecules (Supplementary Fig. 1b). Recombinant TSEN54 showed a high degree of phosphorylation as reported for the endogenous protein (Supplementary Fig. 1c)[36].

Endonuclease activity of tetrameric TSEN was observed in a pre-tRNA cleavage assay using *Saccharomyces cerevisiae* (*S.c.*) pre-tRNA$^{Phe}_{GAA}$ as a substrate, whereas mature *S.c.* tRNA$^{Phe}_{GAA}$ remained unaffected (Fig. 1d). The absence of endonucleolytic activity on mature tRNA confirms the specificity of the complex for its native pre-tRNA substrate. Yeast-two-hybrid experiments with *S.c.* orthologues suggested that strong interactions exist between TSEN15 and TSEN34, as well as between TSEN2 and TSEN54, and that the human endonuclease assembles from preformed dimeric subcomplexes[8]. Using combinatorial co-expression analyses, we identified the formation of stable TSEN15–34 and TSEN2–54 heterodimers (Fig. 1e). Individual heterodimers did not show endonuclease activity, whereas specific endonucleolytic cleavage was observed after stoichiometric mixing of TSEN15–34 and TSEN2–54 in the absence of ATP (Fig. 1e). Size exclusion chromatography confirmed that a stable tetrameric assembly formed upon mixing the individual heterodimers (Supplementary Fig. 1d, e). In line with data from reconstituted TSEN recombinantly produced in *Escherichia coli*[35], these observations indicate that active human TSEN assembles from non-functional, heterodimeric submodules independently of CLP1 and ATP.

**Human TSEN binds precursor and mature tRNAs with similar affinities.** It has been postulated that eukaryotic splicing endonucleases recognize pre-tRNAs via their mature domain[14,15]. To define tRNA binding parameters of human TSEN, we performed interaction studies using catalytically inactive tetramers

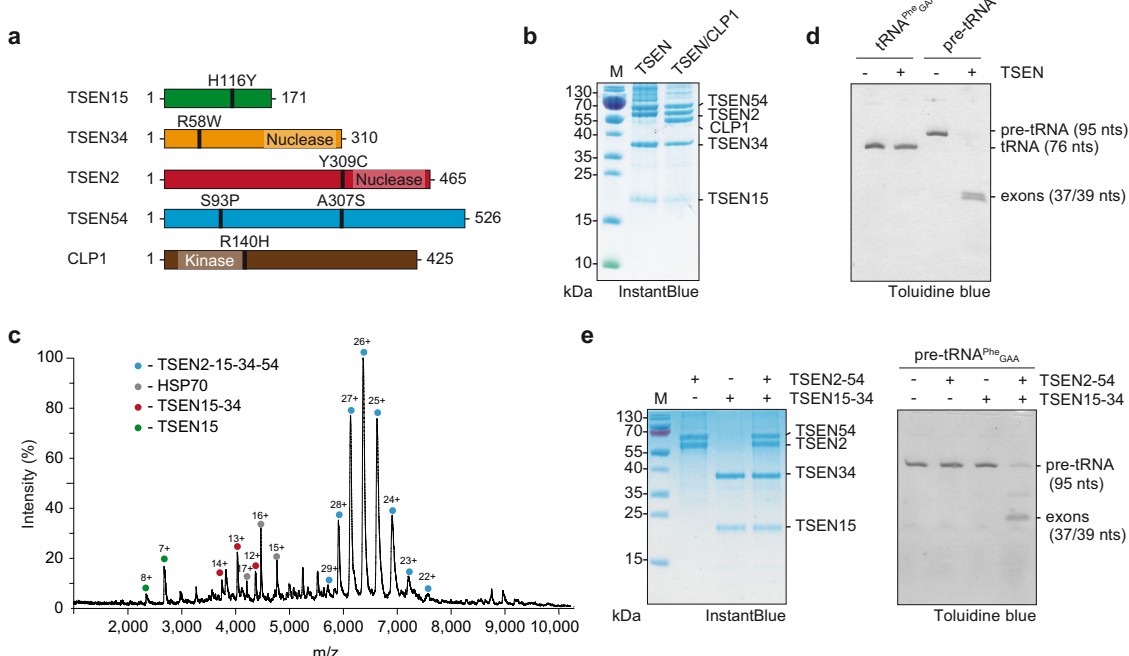

**Fig. 1 Assembly and catalysis of recombinant human TSEN. a** Bar diagrams of TSEN subunits (TSEN15, green; TSEN34, orange; TSEN2, red; TSEN54, blue) and CLP1 (brown) depicting positions of PCH mutations, predicted nuclease domains of TSEN2 and TSEN34, and the RNA kinase domain of CLP1. The total amino acids of each protein are indicated. **b** SDS-PAGE of purified recombinant TSEN and TSEN/CLP1 complexes visualized by InstantBlue staining. Protein identities and size markers are shown. **c** Native mass spectrum of tetrameric TSEN complex from an aqueous ammonium acetate solution. Charge states of the predominant TSEN2–15–34–54 heterotetramer (blue circles), Heat Shock Protein (HSP) 70 (gray circles), the heterodimer TSEN15–34 (red circles), and TSEN15 (green circles) are indicated. **d** Pre-tRNA cleavage assay using tetrameric TSEN complex with *Saccharomyces cerevisiae* (*S.c.*) pre-tRNA$^{Phe}_{GAA}$ and mature tRNA$^{Phe}_{GAA}$. Input samples and cleavage products were separated via urea-PAGE and visualized by Toluidine blue. RNA denominations are given on the right. **e** Pre-tRNA cleavage assay with TSEN heterodimers and *S.c.* pre-tRNA$^{Phe}_{GAA}$. SDS-PAGE of the indicated heterodimers and the reconstituted TSEN tetramer is shown on the left (InstantBlue stain), urea-PAGE of the cleavage products on the right (Toluidine blue stain). Gels are representative of three independent experiments. Source data for **b**, **d**, and **e**, are provided as Source Data files.

(TSEN$^{inactive}$), in which the active site histidines of TSEN2 (His$^{377}$) and TSEN34 (His$^{255}$) were substituted for alanines (Fig. 2 and Supplementary Fig. 2). Alanine substitutions of His$^{377}$ of TSEN2 and His$^{255}$ of TSEN34 abolished cleavage at the 5′ and 3′ splice sites, respectively, and purified TSEN$^{inactive}$ did not cleave pre-tRNA substrates at all (Fig. 2a and Supplementary Fig. 2b).

To perform fluorescence anisotropy and pull-down experiments, we site-specifically labeled precursor and mature yeast tRNA$^{Phe}_{GAA}$ at the terminal 3′ ribose. Despite the inability to cut its native substrate, TSEN$^{inactive}$ interacted stably and specifically with the fluorescently labeled pre-tRNA in a pull-down assay (Fig. 2b). Binding studies using fluorescence anisotropy revealed dissociation constants ($K_D$) of 173 ± 11 nM and 149 ± 20 nM for fluorescently labeled pre-tRNA$^{Phe}_{GAA}$ and mature tRNA$^{Phe}_{GAA}$, respectively (Fig. 2c, d). We determined an inhibition constant ($K_i$) of 197 nM (95% confidence interval of 168–231 nM) in a competition assay confirming the specific interaction, whereas a fluorescent electrophoretic mobility shift assay corroborated a dissociation constant between TSEN and pre-tRNA in the high nanomolar range (Supplementary Fig 2c, d). Our determined $K_D$ values are in good agreement with previously deduced Michaelis constants ($K_M$) of ~30 and 250 nM for intron excision by the yeast and an archaeal tRNA endonuclease, respectively[37]. Taken together, our results show that substrate recognition by human TSEN is primarily mediated by the mature domain of pre-tRNAs which positions intron-containing anticodon stems for cleavage.

**The A–I base pair coordinates cleavage at the 3′ splice site in human TSEN.** Cleavage of archaeal introns strictly relies on the

tRNA BHB motif, whereas the only preserved feature of human tRNA introns is a pyrimidine in the 5′ exon at position −6 from the 5′ splice site which forms a conserved A–I base pair with a purine base at position −3 from the 3′ splice site (Fig. 2e). Studies on the *Xenopus* tRNA endonuclease showed that the A–I base pair is critically involved in the cleavage reaction at the 3′ splice site[20,21,38]. To find out whether the same regulatory principles exist for intron excision in humans, we tested the impact of A–I base pair mutants on endonucleolytic cleavage by tetrameric TSEN (Fig. 2e, f, Supplementary Fig. 2e, and Supplementary Fig. 3). Changing the guanine base G$^{54}$ to cytosine in *S.c.* pre-tRNA$^{Phe}_{GAA}$ produced a pre-tRNA substrate with a disrupted A–I base pair (Fig. 2e, f). Cleavage of this pre-tRNA by wild-type (wt) human TSEN resulted in a 5′ exon and an intron-3′-exon intermediate (Fig. 2f). Cleavage at both splice sites was observed when base pairing at the A–I position was restored by mutating cytosine C$^{32}$ to guanine in the C$^{54}$ background (Fig. 2e, f). The same effect was observed when human pre-tRNA$^{Tyr}_{GTA}$8-1 harboring equivalent mutations were used as substrate (Supplementary Fig. 2e). These findings imply that the presence of an A–I base pair, but not the strict identity of the bases, is essential for cleavage at the 3′ splice site by human TSEN[20,21].

**The molecular architecture of TSEN is evolutionarily conserved.** Our interaction studies using recombinant proteins showed that active human TSEN assembles from inactive TSEN15–34 and TSEN2–54 heterodimers (Fig. 1e). To gain detailed insights into the molecular architecture of the human TSEN complex, we characterized the TSEN15–34 heterodimer by

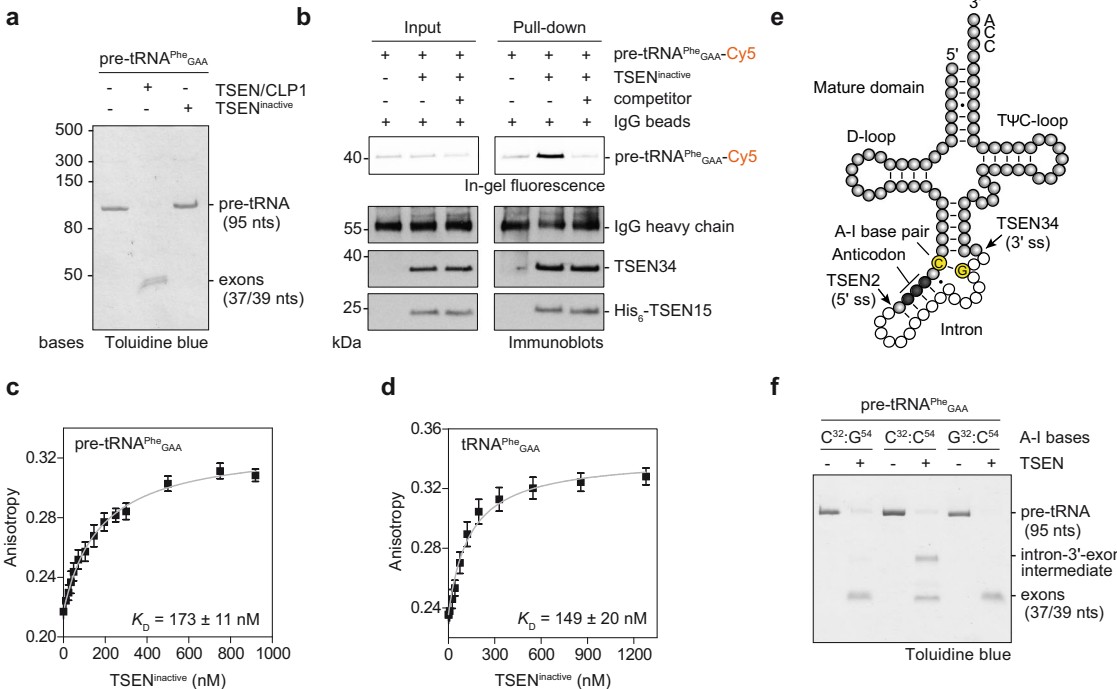

**Fig. 2 Active involvement of the A–I base pair in coordinating pre-tRNA cleavage. a** Pre-tRNA cleavage assay comparing recombinant, inactive TSEN tetramer (TSEN2[H377A] and TSEN34[H255A] double mutant) to the TSEN/CLP1 complex. Cleavage products are visualized by denaturing urea-PAGE with subsequent Toluidine blue staining. RNA size markers are indicated on the left of the gel. **b** Pull-down assay with fluorescently labeled *S.c.* pre-tRNA[Phe][GAA] and inactive, tetrameric TSEN captured on protein G agarose functionalized with an α-His-antibody. Protein size markers are indicated on the left of each immunoblot, protein and RNA identities on the right. Input and co-precipitated, labeled pre-tRNAs were visualized by in-gel fluorescence, TSEN subunits, and the immunoglobulin G (IgG) heavy chain by immunoblotting. The IgG heavy chain served as a loading control. **c** Thermodynamic binding parameters of fluorescently labeled *S.c.* pre-tRNA[Phe][GAA] and inactive, tetrameric TSEN revealed by fluorescence anisotropy. **d** Thermodynamic binding parameters of fluorescently labeled tRNA[Phe][GAA] and inactive, tetrameric TSEN revealed by fluorescence anisotropy. Data in **c** and **d** are presented as mean values ± SD. **e** Schematic depiction of a pre-tRNA molecule showing ribonucleotides belonging to the mature domain (gray spheres), the intronic region (white spheres), the anticodon (black spheres), and the A–I base pair (yellow spheres). Proposed 5′ and 3′ splice sites (ss) are indicated. **f** Impact of A–I base pair mutations in *S.c.* pre-tRNA[Phe][GAA] on the endonucleolytic activity of tetrameric TSEN revealed by a pre-tRNA cleavage assay. C[32]:G[54] – canonical A–I base pair, C[32]:C[54] – disrupted A–I base pair, G[32]:C[54] – inverted A–I base pair. All experiments are representatives of three independent assays. Source data for **a**, **b**, and **f**, are provided as Source Data files.

X-ray crystallography (Fig. 3 and Supplementary Fig. 4). Despite extensive crystallization trials, full-length TSEN15–34 did not crystallize. To define a crystallizable core complex, we subjected the TSEN15–34 complex to limited proteolysis with subsequent size exclusion chromatography and MS analysis (Supplementary Fig. 4a, b and Supplementary Tables 2–4). We observed two comigrating polypeptide species corresponding to residues 23–170 of TSEN15 and residues 208–310 of TSEN34 covering the predicted conserved nuclease domains (Supplementary Fig. 4b, c and Supplementary Fig. 5)[8].

We re-cloned, co-expressed and purified the proteolytically characterized fragments, which readily formed rod-shaped crystals in space group P2₁ and diffracted X-rays to a resolution of 2.1 Å (Supplementary Fig. 4d and Supplementary Table 5). The asymmetric unit is composed of two domain-swapped TSEN34 molecules, each binding one TSEN15 protomer at their C-terminal domains (Supplementary Fig. 4e, f). The domain swap is brought about by a short, structured N-terminal α-helix/ β-hairpin element of TSEN34 that is liberated to hook onto the neighboring protomer, presumably due to the truncated N-terminus of the molecule. The domain swap is most likely a crystallization artifact, since TSEN15–34 migrates as a hetero-dimer during size exclusion chromatography (Supplementary Fig. 4d) and forms dimers of dimers at high protein concentration as shown by size exclusion chromatography multi-angle light scattering (SEC-MALS) (Supplementary Fig. 4g). The two

TSEN15 and the two TSEN34 molecules in the asymmetric unit are very similar with average overall RMSDs of 0.37 and 0.50 Å, respectively. In one TSEN15 protomer, an elongated C-terminal region (residues 162–170) is visible, which is stabilized by crystal contacts (Supplementary Fig. 4f).

TSEN15 and TSEN34 display the typical endonuclease fold with the latter harboring the Tyr[247]/His[255]/Lys[286] catalytic triad as also found in archaeal and eukaryotic endonucleases (Fig. 3a, b and Supplementary Fig. 4h)[13]. The dimeric TSEN15–34 complex is characterized by an elongated central twisted β-sheet connected by the C-terminal β-strands of TSEN15 and TSEN34 with a buried surface area of ~1980 Å[2] between the protomers (Fig. 3a, b). Each face of the individual twisted β-sheets of TSEN15 and TSEN34 is mainly stabilized by hydrophobic interactions to an alpha helix (Fig. 3a, c, d). In the interface between TSEN15 and TSEN34, two structural water molecules are found, which are coordinated by hydrogen bonds to backbone oxygens or amide groups of Ile[110], Ala[112], and Leu[114] of TSEN15, Ile[269], Leu[271], and Gln[272] of TSEN34, and the side-chain oxygen of Gln[272] (Fig. 3c). Furthermore, a YY motif in TSEN15 (Tyr[152]/Tyr[153]), which is conserved in eukaryotic endonucleases and archaeal α₄- and (αβ)₂-type endonucleases (Supplementary Fig. 6) both stabilizes TSEN15 by hydrophobic interactions and the dimer interface by hydrogen bonds to the main-chain carbonyl oxygen of Leu[274] and the side-chain oxygen of Ser[283] of TSEN34 (Fig. 3a, c). In contrast to a previous solution NMR structure of homodimeric TSEN15[9], our biochemical and structural

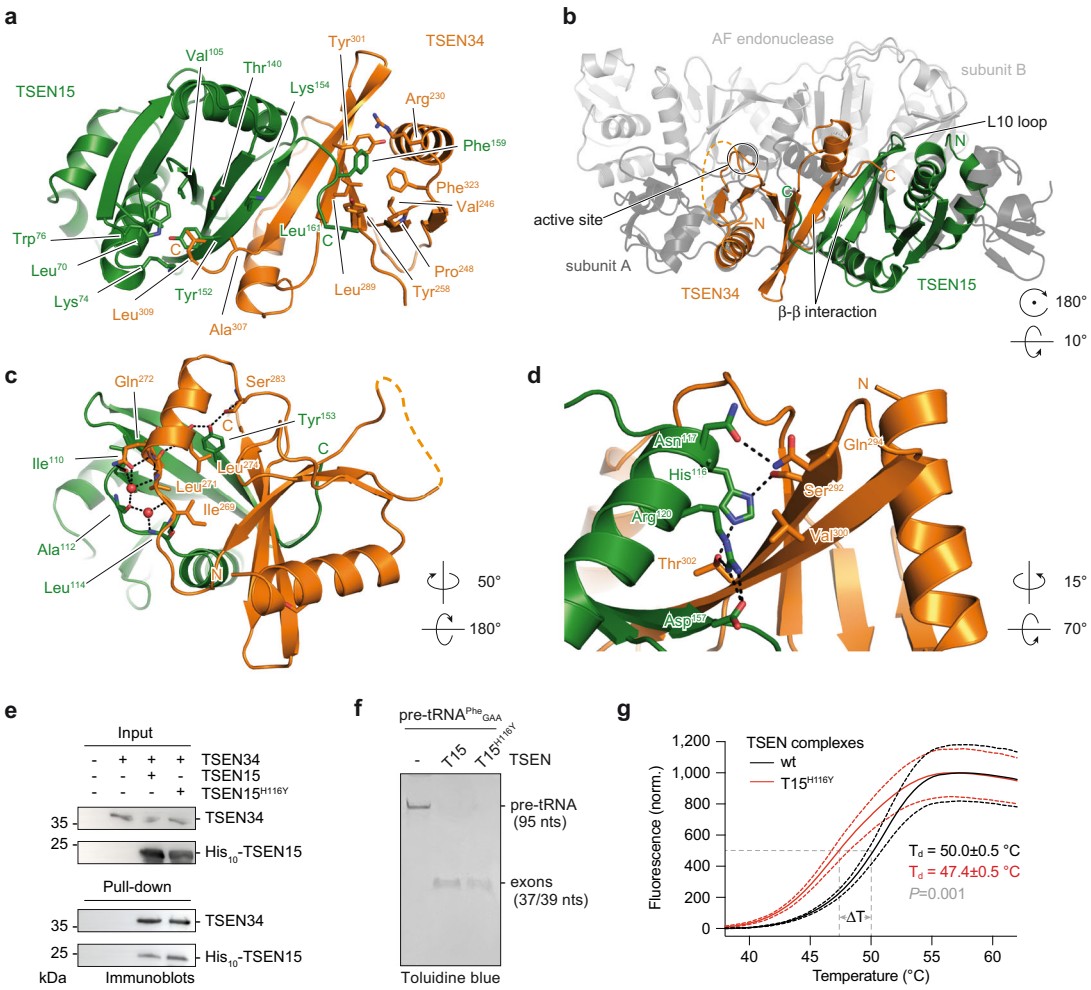

**Fig. 3 Structural and functional details of the TSEN15–34 dimer interface. a** X-ray crystal structure of a TSEN15–34 complex derived from limited proteolysis experiments. TSEN15 (green) and TSEN34 (orange) are shown in cartoon representation. Key amino acids are depicted in stick representation together with amino- (N) and carboxy (C)-termini. **b** Superposition of the TSEN15–34 heterodimer and the pre-tRNA endonuclease from *Archaeoglobus fulgidus* (AF) (PDB ID 2GJW)[13]. The position of the catalytic triad of TSEN34 (active site), the L10 loop of TSEN15, and the β-strands involved in the β-β interaction between TSEN15 and TSEN34 are shown. **c** Cartoon representation of the dimer interface with amino acid residues in stick representation (color coding as in **a**). Water molecules and hydrogen bonds are shown as red spheres and black dashed lines, respectively. **d** Cartoon representation of the TSEN15–34 interface highlighting histidine 116 (His[116]) of TSEN15, mutated in patients with a PCH type 2 phenotype. **e** Pull-down experiment with TSEN34, wt TSEN15, and TSEN15 carrying the H116Y mutation. Input and co-precipitated proteins were separated by SDS-PAGE and visualized by immunoblotting. Size markers and protein identities are shown. Blots are representative of two independent experiments. **f** Pre-tRNA cleavage assay with wt, tetrameric TSEN complex, and a tetrameric TSEN complex carrying the TSEN15[H116Y] (T15[H116Y]) mutation. Cleavage products were separated by urea-PAGE and visualized with toluidine blue. Experiments are representative of two independent assays. **g** Thermal stability of wt, tetrameric TSEN (black line), and TSEN15[H116Y] mutant complex (red line) assessed by differential scanning fluorimetry (DSF). Note that recombinant complexes were purified from HEK293 cells. Normalized (norm.) fluorescence is plotted against temperature in degree Celsius (°C). Denaturation temperatures ($T_d$) of the complexes were derived from sigmoidal Boltzmann fits (gray dashed lines) with an error of fits. Standard deviations (SD) of technical triplicates are represented by red (T15[H116Y]) and black (wt) dashed lines ($P = 0.001$; unpaired, two-tailed Student´s *t*-test). Source data for **e** and **f** are provided as Source Data files.

analyses show that the assembly and architecture of TSEN are conserved and support the hypothesis that tRNA splicing endonucleases arose from a common ancestor through gene duplication and differentiation events[39].

**PCH-causing mutations destabilize recombinant TSEN.** A previous genetic study identified a His-to-Tyr substitution at position His[116] of TSEN15 in patients with PCH type 2 (Fig. 3d)[30]. The imidazole group of His[116] is central to a hydrogen bond network involving Asn[117], Arg[120], and Asp[157] of TSEN15 and Ser[292] and Thr[302] of TSEN34 (Fig. 3d). We tested the impact of this substitution in a pull-down experiment using full-length TSEN15 and TSEN34 and also in a pre-tRNA cleavage assay in the context of the tetrameric assembly (Fig. 3e, f). We hypothesized that the

substitution impairs complex formation and activity due to steric clashes in the dimer interface and loss of the hydrogen bond network. However, TSEN15 carrying the His-to-Tyr mutation engaged in complex formation with TSEN34 similar to the wt protein, and no impairment of catalytic activity was observed (Fig. 3e, f). We assumed that the large hydrophobic interface compensates for the loss of the hydrogen bond network. To assess the effects of the TSEN15[H116Y] mutation on the thermal stability of TSEN, we compared the mutant complex to wt by differential scanning fluorimetry (DSF, Fig. 3g)[40]. This assay reported apparent denaturing temperatures of 50.0 ± 0.5 °C and 47.4 ± 0.5 °C for wt and mutant TSEN, respectively (Fig. 3g and Supplementary Fig. 4i). These data suggest that destabilization of TSEN might be a general effect of PCH-causing mutations.

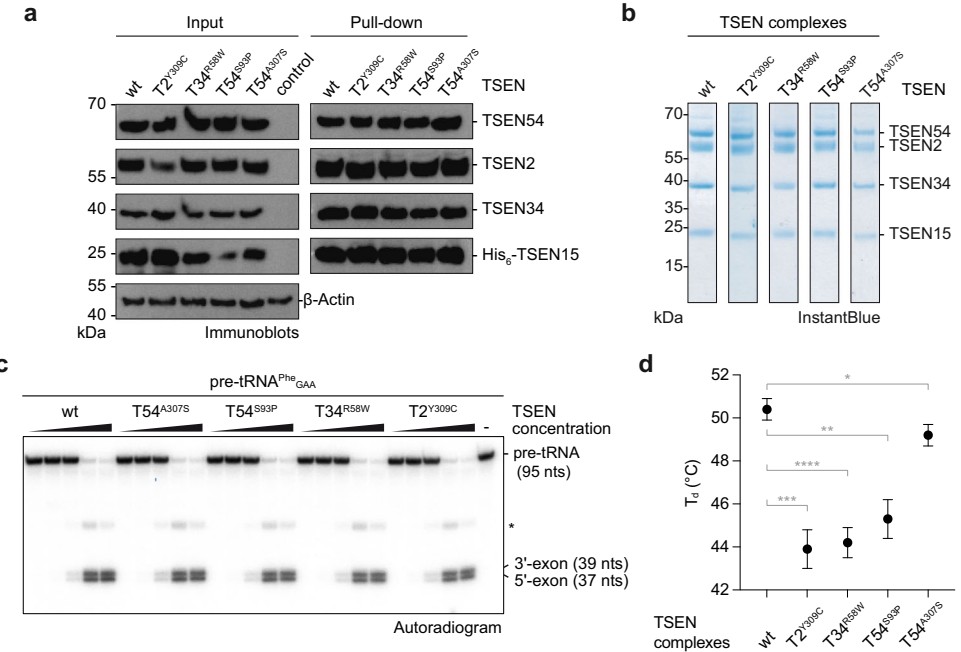

**Fig. 4 PCH mutations affect thermal stability but not activity of recombinant TSEN. a** Immunoblot analysis of a pull-down assay with wt and mutant TSEN complexes co-expressed in HEK293 cells. Size markers and protein identities are indicated. **b** SDS-PAGE of SEC peak fractions of purified recombinant wt and mutant heterotetrameric TSEN complexes. **c** Pre-tRNA endonuclease assay of radioactively labeled *S.c.* pre-tRNA$^{Phe}_{GAA}$ with increasing amounts of recombinant wt or mutant TSEN complexes revealed by phosphorimaging. **d** Thermal stability of recombinant wt and mutant TSEN complexes assessed by DSF. Shown are means of the denaturation temperature ($T_d$) of each complex with fit errors derived from Boltzmann sigmoids. Fit errors were derived from means of technical triplicates and are representative of biological duplicates (*$P = 0.0424$, **$P = 0.0010$, ***$P = 0.0004$, ****$P = 0.0002$; unpaired two-tailed Student's *t*-test). Source data for **a**, **b**, and **c**, are provided as Source Data files.

It is not known how mutations in TSEN subunits lead to disease, given that the complex is essential for cell physiology and survival and yet only a subset of specific cell types is impacted[31]. It has been suggested that PCH-associated mutations interfere with complex assembly, stability, or enzymatic activity leading to a general decrease of TSEN functionality which is eventually exacerbated in specific cellular or tissue contexts. Given that the His-to-Tyr mutation in TSEN15 thermally destabilized the endonuclease complex, we produced heterotetrameric TSEN complexes carrying the PCH-causing mutations TSEN2$^{Y309C}$, TSEN34$^{R58W}$, TSEN54$^{S93P}$, and TSEN54$^{A307S}$ in HEK293 cells and performed pull-down experiments to assess complex assembly and integrity (Fig. 4a). Despite subtle differences in expression levels of the individual subunits, pull-down via TSEN15 co-precipitated TSEN2, TSEN34, and TSEN54, irrespective of the introduced PCH-causing mutation (Fig. 4a). Control pull-downs from HEK293 cells overexpressing only His-tagged TSEN15 showed that endogenous subunits of TSEN do not associate with overexpressed TSEN15, probably due to their very low copy numbers (Supplementary Fig. 7a). We produced and purified recombinant heterotetrameric TSEN complexes carrying the pathogenic missense mutations from baculovirus-infected insect cells (Fig. 4b) and also did not see obvious deleterious effects on subunit composition or pre-tRNA cleavage kinetics (Fig. 4b, c and Supplementary Fig. 7b).

Considering the low abundance of TSEN molecules in cells[5,8] and that PCH mutations phenotypically affect only cerebellar neurons, we reasoned that expression levels in our reconstitution systems might be too high to reveal subtle alterations in TSEN assembly and function. To assess the effects of PCH-causing mutations on complex stability, we used the DSF assay (Fig. 3g, Supplementary Fig. 4i, and Supplementary Fig. 7c). Most PCH-causing mutations led to substantial shifts towards lower denaturation temperatures (*e.g.* $T_d$ of $43.9 \pm 0.9$ °C for TSEN2$^{Y309C}$ compared to $T_d$ of $50.4 \pm 0.5$ °C for wt TSEN) when exposed to thermal gradients indicating that mutant TSEN complexes have compromised structural integrity (Fig. 4d and Supplementary Fig. 7c). The relative changes in thermostability ($T_\Delta$) of the mutant complexes compared to wt TSEN ranged from 6.5 °C for the TSEN2$^{Y309C}$ mutation to 1.2 °C for the TSEN54$^{A307S}$ mutation and might explain different severities of disease phenotypes (Fig. 4d and Supplementary Fig. 7c)[26]. DSF data also revealed two-state unfolding behaviors for all TSEN complexes when analyzed by the ProteoPlex algorithm[41] suggesting cooperativity of unfolding transitions for the individual subunits (Supplementary Table 6), thus explaining why mutations in different subunits lead to an overall destabilization of TSEN. Our data suggest that PCH phenotypes in patients potentially develop due to destabilized TSEN complexes.

**Pre-tRNA processing is impaired in PCH patient cells**. To determine if the pre-tRNA processing activity is compromised in PCH patients we derived primary skin fibroblasts from PCH patients, their healthy parents (when available), and unrelated controls (Supplementary Table 7). We chose the common *TSEN54* c.919 G >T (TSEN54$^{A307S}$) mutation, which is reported in ~90% of recognized TSEN-linked PCH cases[31,32] and for which a large cohort of patient samples is available. The cell lines we created did not show any morphological differences compared to control cells. However, when we assayed lysates derived from homozygous *TSEN54* c.919 G >T cell lines, we observed a drastic reduction in pre-tRNA splicing efficiency compared to control cell lysates (Fig. 5a). Subtle differences in ligation efficiency, as observed for cell line Ba2, may result from the fibroblasts having different genetic backgrounds.

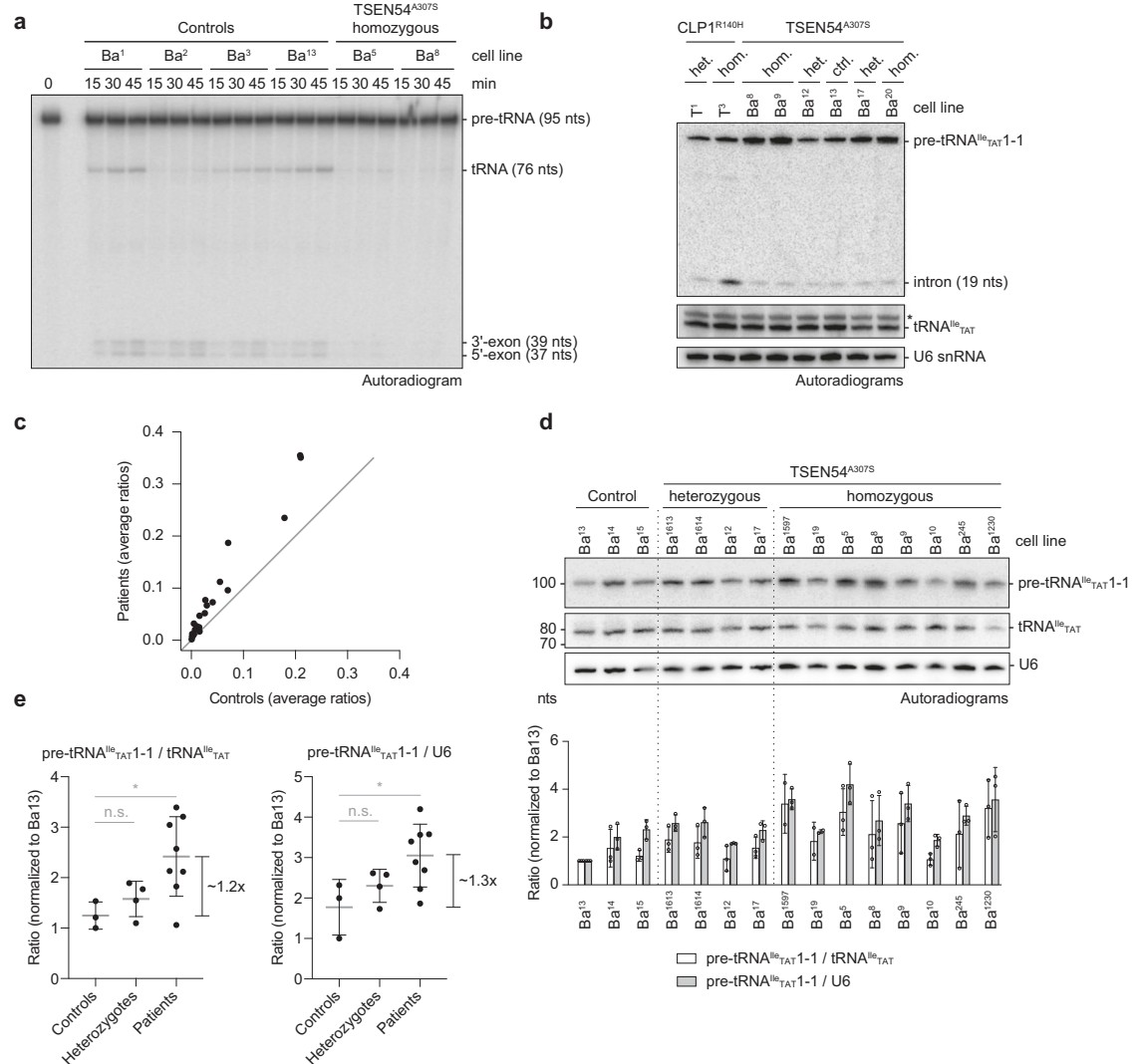

**Fig. 5 _TSEN54_ c.919 G >T fibroblasts exhibit reduced splicing activity in vitro and accumulation of intron-containing pre-tRNAs. a** Pre-tRNA splicing assay (time course) with radioactively labeled _S.c._ pre-tRNA[Phe]GAA and cell extracts derived from control and PCH patient fibroblasts. Splicing products were separated by urea-PAGE and monitored by phosphorimaging. **b** Comparison of pre-tRNA[Ile]TAT1-1 intron abundance between control cells and fibroblasts carrying the heterozygous or homozygous _TSEN54 c.919_ G >T (TSEN54[A307S]) or the _CLP1 c.419_ G > A (CLP1[R140H]) mutation by northern blotting. **c** Average ratios of pre-tRNAs to mature tRNAs derived from Hydro-tRNAseq for all intron-containing tRNAs comparing PCH patients to wild-type control samples. The black line indicates a slope of 1. **d** Northern blot analysis comparing pre-tRNA[Ile]TAT1-1 levels to levels of mature tRNA[Ile]TAT or U6 snRNA in control fibroblasts and fibroblasts carrying the heterozygous (het.) or homozygous (hom.) _TSEN54 c.919_ G > T mutation. The data are representative of three independent experiments. Signal intensities were quantified and displayed as ratios normalized to Ba[13] in the bottom panel. Data are presented as mean values ± SD. **e** Statistical representation of northern blot analysis in d. Mean ratios of signal intensities for pre-tRNA[Ile]TAT1-1 to tRNA[Ile]TAT (left panel) or to U6 snRNA (right panel) were normalized to Ba[13], and grouped into control, heterozygous, and homozygous patient classes (n = 3, different control fibroblast cell lines; n = 4, different heterozygous fibroblast cell lines; n = 8, different homozygous PCH patient fibroblast cell lines carrying the _TSEN54_ c.919 G > T mutation). Unpaired Student´s _t_-test (two-tailed) for ratios of pre-tRNA[Ile]TAT1-1 to mature tRNA[Ile]TAT or to U6 snRNA revealed a significant difference between control and patient cell lines of 1.2-fold (*P = 0.0371) or 1.3-fold (*P = 0.0344), respectively. Data are presented as mean values ± SD. Panels in a and b are representative of at least two independent experiments. Source data for **a**, **b**, **c**, and **e**, are provided as Source Data files.

This result is reminiscent of our observations in patient-derived cell lines carrying a homozygous _CLP1_ c.419 G >A (CLP1[R140H]) mutation, which leads to severe motor-sensory defects, cortical dysgenesis, and microcephaly[24,25]. In contrast to homozygous CLP1[R140H] cells, in which introns accumulate, intron accumulation did not occur in either heterozygous or homozygous TSEN54[A307S] backgrounds as judged by northern blot analysis using a probe specific for the intron of pre-tRNA[Ile]TAT1-1 (Fig. 5b). These results suggest an impairment in intron excision rather than a defect in downstream processes of

the tRNA splicing reaction, which might lead to the accumulation of pre-tRNAs in patient cells.

To test this hypothesis, we compared levels of intron-containing pre-tRNAs in cell lines carrying the homozygous _TSEN54_ c.919 G >T mutation to heterozygous cell lines and controls by hydro-tRNAseq[4] (Fig. 5c and Supplementary Fig. 8). Despite some variation among cell lines of the same TSEN genotype, we observed an accumulation (~2–6 fold) of intron-containing pre-tRNAs in homozygous _TSEN54_ c.919 G >T cell lines compared to control cell lines, albeit global levels of mature

tRNAs remained largely unaffected (Fig. 5c and Supplementary Fig. 8). The distributions of the ratios of precursor over mature tRNA reads showed that there was no bias for the enrichment of any specific precursor tRNA among samples (Supplementary Fig. 8a). The difference in ratio medians between homozygous *TSEN54* c.919 G >T and control cell lines was statistically significant (*P* <0.0001, two-tailed paired Wilcoxon signed-rank test). We also analyzed pre-tRNA levels by northern blotting and found that *TSEN54* c.919 G >T cells indeed show higher levels of intron-containing pre-tRNAs than control cells (Fig. 5d, e), although by lower margins than those observed by hydro-tRNAseq. We measured by northern blot analysis a 1.2- or 1.3-fold increase of pre-tRNA$^{Ile}_{TAT}$1-1 levels, standardized to mature tRNA$^{Ile}_{TAT}$ or U6 snRNA, respectively, and a ~2–6-fold increase by hydro-tRNAseq. Thus, despite differences in the sensitivity of the two techniques, we conclude that the *TSEN54* c.919 G >T mutation results in an increase of the steady-state levels of intron-containing tRNAs in PCH patients fibroblasts. Furthermore, our data indicate that defects in pre-tRNA processing by mutated TSEN are unrelated from those involving mutated CLP1.

**Pre-tRNA processing defects are linked to altered TSEN stability.** To investigate whether the reduction of pre-tRNA processing in cell extracts of homozygous *TSEN54* c.919 G >T patients was due to altered TSEN assembly or stability, we performed quantification and co-immunoprecipitation experiments of endogenous TSEN using rabbit polyclonal antibodies raised against peptides of TSEN2 (α-TSEN2), TSEN34 (α-TSEN34), and TSEN54 (α-TSEN54)[36]. Immunoblot analyses showed that the homozygous *TSEN54* c.919 G >T mutation does not impact steady-state levels of TSEN54, suggesting that no changes in either mRNA stability, transcription rate, or protein turnover occur (Fig. 6a).

To evaluate TSEN complex composition and pre-tRNA cleavage activity, we performed immunoprecipitation experiments from patient-derived and control fibroblasts using α-TSEN2 or α-TSEN34 antibodies (Fig. 6b, c and Supplementary Fig. 9). Immunoprecipitation using an α-TSEN34 antibody showed a substantial reduction of co-immunoprecipitated TSEN2 and TSEN54 from patient cell lines (Fig. 6b and Supplementary Fig. 9a, b), while at the same time pre-tRNA cleavage activity was strongly diminished in α-TSEN2 and α-TSEN34 immunoprecipitates (Fig. 6c and Supplementary Fig. 9c, d). These results indicate that TSEN assembly defects lead to reduced pre-tRNA cleavage in PCH patient cells. Since the association of TSEN2 and TSEN54 is likewise affected but steady-state levels of the individual proteins are not, we conclude that lower TSEN activity is a consequence of altered complex formation in patient cells.

## Discussion

Here we report the recombinant expression, purification, and assembly of functional human TSEN/CLP1 complex, and reveal that TSEN stability and activity are perturbed in PCH patient cells. We show that heterotetrameric TSEN is assembled from heterodimeric TSEN15–34 and TSEN2–54 subcomplexes, which combine to form the composite active sites for catalysis (Fig. 1e). The nuclease fold seen in our TSEN15–34 X-ray crystal structure is conserved (Fig. 3b)[13,16] suggesting that the TSEN2–54 heterodimer — and the entire TSEN complex — likely forms through interactions similar to those seen in the TSEN15–34 heterodimer, as well as related interactions previously observed in archaeal tRNA endonucleases. Our interaction studies with catalytically inactive TSEN mutants show that substrate recognition occurs through interactions with the mature tRNA fold, including the aminoacyl acceptor stem, the D-loop, and the Ψ-loop, and

support the ruler model of substrate recognition (Fig. 2c, d)[8]. The similar affinities TSEN shows for pre-tRNAs and tRNAs suggest that thermodynamic effects are unlikely to play a role in substrate selection (Fig. 2c, d). Instead, we speculate that different binding kinetics should contribute to the selection of pre-tRNAs over mature tRNAs, thereby ensuring efficient scanning and processing of the large pre-tRNA pool. Alternatively, a higher concentration of intron-containing pre-tRNAs in the nucleus might contribute to TSEN substrate specificity. TSEN was shown to cleave artificial intron-containing anticodon stem-loop structures[35]. A three-dimensional structure of TSEN with substrate RNA will help define how eukaryotic TSEN recognizes pre-tRNAs and anticodon stem-loop structures in particular[35,42].

The tRNA splicing machinery is involved in the processing of other RNA species[43–46]. Eukaryotic tRNA endonucleases are involved in the processing of mRNAs and rRNAs[5,43,47,48]. TSEN and CLP1 are key factors in the generation of tRNA intronic circular (tric) RNAs, a poorly characterized class of short non-coding RNAs in *Drosophila* and humans[35,46]. Archaeal tRNA endonucleases are capable of binding and cutting any RNA fragment that adopts a BHB motif[19]. tRNA splicing in *Xenopus* necessitates a purine/pyrimidine base pair at the A–I base pair positions for 3′ splice site recognition and cleavage. Our data show that requirements for cleavage at the 3′ splice site by human TSEN are more relaxed and only need the A–I base pair, whereas the purine/pyrimidine identities of the bases are negligible (Fig. 2f and Supplementary Fig. 2e). The relaxed specificity may facilitate the recognition and cleavage of non-canonical substrates. However, structures of human tRNA endonucleases with bound pre-tRNA substrate confirming this hypothesis are still missing. Nonetheless, our data suggest that substrate recognition and cleavage by human TSEN are two distinct processes with different structural requirements regarding the RNA.

While we show that the assembly and enzymatic function of recombinant human TSEN complexes are not hampered by PCH-associated mutations, these mutations cause thermal destabilization in vitro and affect complex assembly and activity in patient cells. Structural studies on archaeal tRNA endonucleases show that there are two major interaction interfaces: the β-β-interaction, mainly driven by hydrophobic interactions, and the L10 loop, involving hydrogen bonds and salt bridges. The hydrophobic interface has a higher degree of plasticity and thereby could accommodate mutations to a certain extent, whereas interactions within the hydrophilic interface are less tolerant to changes. Given the low estimated abundance of TSEN subunits (~100 molecules per cell)[7], destabilization by PCH-associated mutations may have a strong effect on the assembly of the heterotetramer, whereas the individual heterodimers might be sufficiently stable to escape protein degradation. In line with this hypothesis, we find decreased levels of TSEN2 and TSEN54 in α-TSEN34 immunoprecipitates from PCH patient cells (Fig. 6b and Supplementary Fig. 9a).

By investigating TSEN activity in patient fibroblasts, we obtained evidence that the TSEN54$^{A307S}$ mutation significantly impacts complex stability and pre-tRNA splicing efficiency. We detected only a modest increase of pre-tRNA levels, indicating that residual TSEN activity is sufficient to sustain the necessary tRNA production, in agreement with the fact that these cells do not exhibit any obvious phenotype. These observations are likely to extend also to other PCH-related, mutation-containing complexes, for which we measured an even larger reduction of thermal stability.

How TSEN mutations lead to a disease phenotype only in a subset of neurons, resulting in the selective degeneration of cerebellar and – to a variable extent – anterior cortical structures, is not understood[27,49]. Our results demonstrate that these

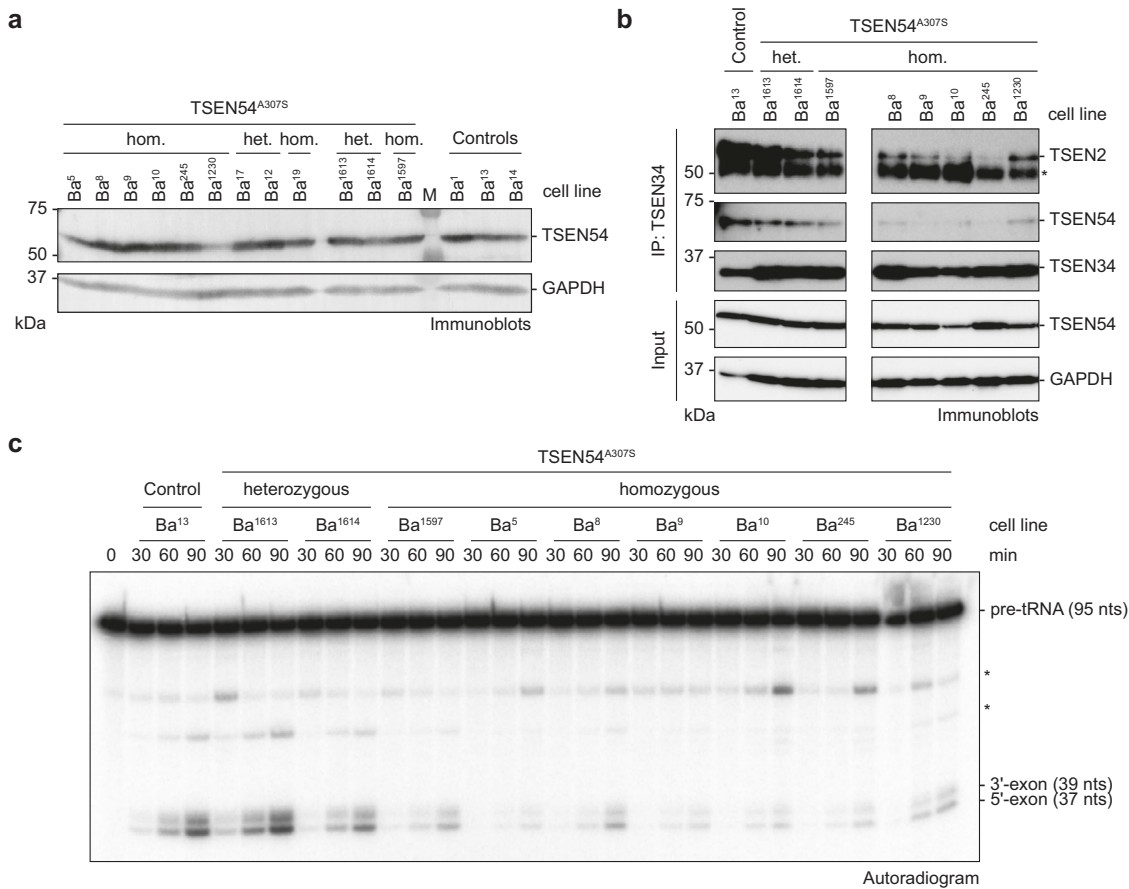

**Fig. 6 Reduced pre-tRNA cleavage activity in PCH patient-derived cell extracts is associated with altered composition of TSEN. a** Comparison of TSEN54 protein levels between control and heterozygous or homozygous PCH patient fibroblasts analyzed by immunoblotting. GAPDH served as a loading control. M, protein size marker. **b** Co-immunoprecipitation (IP) assay using an α-TSEN34 antibody with cell lysates derived from control and heterozygous or homozygous *TSEN54 c.919* G > T fibroblasts analyzed by immunoblotting. The asterisk indicates the heavy chain of the α-TSEN34 antibody. **c** On-bead pre-tRNA cleavage assay (time course) with radioactively labeled *S.c.* pre-tRNA$^{Phe}_{GAA}$ and immunoprecipitated TSEN complexes (α-TSEN34 antibody-coupled resin) derived from control fibroblasts and from fibroblasts carrying heterozygous or homozygous *TSEN54 c.919 G > T* mutation shown in (**b**). Unspecific bands are indicated by asterisks. Data are representative of at least two independent experiments. Source data for **a**, **b**, and **c**, are provided as Source Data files.

mutations cause a significant complex destabilization in vitro and, at least in the case of TSEN54$^{A307S}$, in patient fibroblasts. A reduction in TSEN activity, tolerated in most cell types, could have dramatic consequences in neurons. Such a model has already been proposed to explain the observation that defects in many mRNA and tRNA processing pathways often result in various types of neuronal diseases[50–52].

Several factors could potentially act as neuron-specific triggers of the disease. For instance, although an adequate supply of tRNAs is essential for protein biosynthesis in all cell types[50], neurons may be particularly susceptible to subtle translation defects and, consequently, defects in proteostasis[53]. Neurons require rapid and localized protein synthesis for synaptic plasticity, which needs coordinated transport of the translational machinery, mRNAs, and tRNAs themselves. Changes in tRNA levels could influence the local speed of mRNA translation in a tissue-specific manner depending on the availability of cognate tRNAs[54]. Balanced kinetics of tRNA accumulation could be crucial in tissues or cell subpopulations with a high metabolism, so that an otherwise modest defect in production rate might be deleterious where there is a temporally and spatially restricted high demand[55,56].

Another possibility is that impaired TSEN function may selectively impact the processing of cerebellum-specific pre-

tRNAs. It has been shown that changes in tRNA repertoires correlate with the codon usage of genes implicated in cell proliferation or differentiation, to fine-tune their translation[57–59]. In mammals, expression of tRNA isodecoder families (tRNAs with the same anticodon but sequence differences in the tRNA body) varies among tissues and during development and can be altered under disease conditions[57,60–62]. In this scenario, neuron-specific isodecoders could be critically reduced in PCH patients, as a result of TSEN failure to cleave specific precursors. Such event has for instance been described in mouse, where a mutation in a tRNA gene specifically expressed in the central nervous system exhibits a synthetic effect with the loss of a ribosome recycling factor, selectively inducing cerebellar neurodegeneration of cerebellar granules, without affecting other cell types[63]. In a similar scenario, neuron-specific isodecoders could be critically reduced in PCH patients, due to TSEN failure to cleave specific precursors. A secondary, potentially deleterious consequence of this failure could be the aberrant accumulation of pre-tRNAs, at levels much higher than those we measured in fibroblasts.

tRNAs also function as signaling molecules in the regulation of numerous metabolic and cellular processes, or as stress sensors and in tRNA-dependent biosynthetic pathways[64]. Transfer RNA-derived fragments (tRFs) have been identified as small non-coding RNAs contributing to translational control, gene

regulation, and silencing, as well as progressive motor neuron loss[65,66]. Therefore, in another potential mechanism leading to PCH, impaired TSEN activity could result in unbalanced tRF levels, with deleterious effects on cell physiology.

In summary, our data link a pre-tRNA splicing defect to PCH, but additional factors and cellular mechanisms are likely to be involved in the development of the disease. Splicing of pre-tRNAs may require spatial regulation and local confinement. In this regard, altered complex stability might affect interactions between TSEN, CLP1, or other cellular components. For instance, TSEN activity has been linked to mRNA processing in yeast[45,47,48]. Thus, certain neuron-specific mRNA transcripts might require some so-far-uncharacterized functions of TSEN and CLP1, which are impaired by the disease mutations. Clearly, future studies will be needed to address these questions in vivo and to build disease models.

## Methods

**Plasmid constructs**. To enable recombinant protein production in insect and mammalian cells using a single set of transfer vectors, we modified the MultiBac expression vector suite[33,67] by replacing existing promoters with a dual CMV-p10 promoter[34] to derive the acceptor vector pAMI, and the three donor vectors pMIDC, pMIDK, and pMIDS (Supplementary Fig. 1). Open reading frames encoding the TSEN subunits TSEN2 (UniProtKB Q8NCE0), TSEN15 (UniProtKB Q8WW01), TSEN34 (UniProtKB Q9BSV6), TSEN54 (UniProtKB Q7Z6J9), and CLP1 (UniProtKB Q92989) were amplified by polymerase chain reaction (PCR) and cloned into the modified MultiBac vectors leading to pAMI_CLP1, pAMI_TSEN2, pMIDC_TSEN54, pMIDK_TSEN15, and pMIDS_TSEN34. An N-terminal His$_6$-tag followed by a Tobacco Etch Virus (TEV) protease cleavage site was engineered in vectors encoding CLP1, TSEN2, and TSEN15, leading to pAMI_His$_6$-TEV-CLP1, pAMI_His$_6$-TEV-TSEN2, and pMIDK_His$_6$-TEV-TSEN15, respectively. Furthermore, a pMIDK plasmid encoding TSEN15 with an N-terminal TEV protease-cleavable Streptavidin-binding peptide (SBP) tag was generated (pMIDK_SBP-TEV-TSEN15). The PCH-causing mutations Tyr$^{309}$Cys (TSEN2), His$^{116}$Tyr (TSEN15), Arg$^{58}$Trp (TSEN54), Ser$^{93}$Pro (TSEN54), and Ala$^{307}$Ser (TSEN54), and the active site mutations His$^{255}$Ala (TSEN34), and His$^{377}$Ala (TSEN2) were introduced via QuikChange mutagenesis. For crystallographic purposes, the coding sequences of TSEN34 (residues 208–310) and TSEN15 (residues 23–170) were cloned into pAMI and pMIDK, respectively, attaching an N-terminal His$_{10}$-tag followed by a TEV protease cleavage site to TSEN15. Prior to integration into the EMBacY baculoviral genome via Tn7 transposition[67], acceptor and donor vectors were concatenated by Cre-mediated recombination utilizing the LoxP sites present on each vector. For co-expression of the TSEN15–34 heterodimer, the vectors pMIDK_His$_6$-TEV-TSEN15 and pMIDS_TSEN34 were concatenated with the vector pADummy, which was generated by removing the CMV-p10-SV40 expression cassette from pAMI by cleavage with AvrII and SpeI restriction enzymes and re-ligation of the backbone.

For two-color pre-tRNA cleavage assays, TSEN/CLP1-FLAG and TSEN-STREP wt complexes were cloned into pBIG2ab and pBIG1a expression vectors, respectively, using the biGBac cloning system[68]. ORFs for each TSEN subunit were synthesised into pUC57 vectors and then cloned using BamHI and HindIII into pLIB. TSEN2$^{H377A}$ and TSEN34$^{H255A}$ point mutants were generated using the Q5 site-directed mutagenesis kit (New England Biolabs) prior to assembly into biGBac vectors, generating both the TSEN/CLP1-FLAG (TSEN2$^{H377A}$) and TSEN/CLP1-FLAG (TSEN34$^{H255A}$) pBIG2ab constructs.

Yeast and human pre-tRNA genes were amplified by PCR from genomic DNA of Saccharomyces cerevisiae strain S288C and human embryonic kidney (HEK293) cells, respectively. Pre-tRNA sequences were optimized for in vitro transcription (GG at the starting position, CC at pairing position in acceptor stem) and flanked by a preceding T7 promoter sequence and a BstNI cleavage site at the 3′ end of each pre-tRNA. DNA fragments were cloned into a pUC19 vector via sticky end ligation using BamHI and HindIII restriction sites. Mature tRNA sequences were obtained by deleting the intron sequence using the Q5 Site-Directed Mutagenesis kit (New England Biolabs). All constructs in this study were verified by Sanger sequencing. A list of all primers used in this study is provided in Supplementary Table 8.

**Production and purification of human TSEN complexes**. Recombinant human TSEN complexes were overexpressed in Spodoptera frugiperda (Sf) 21 cells essentially as described[33,67,69]. In brief, transfer plasmids encoding TSEN subunits were created by Cre-mediated recombination and recombinant baculoviral BACs were generated by Tn7 transposition in Escherichia coli DH10EMBacY cells (Geneva Biotech). Sf21 cells were grown in Sf-900 II SFM medium (Thermo Fischer Scientific), transfected with recombinant EMBacY BACs using X-tremeGENE DNA Transfection Reagent (Roche), and incubated for 72 h at 28 °C. Recombinant initial baculoviruses (V$_0$) were harvested from cell supernatants and used for the production of amplified baculovirus (V$_1$) in

Sf21 suspension cultures at a multiplicity of infection (MOI) <1. Typically, TSEN complexes were produced in 1.6 liters of Sf21 suspension culture at a cell density of $1 \times 10^6$ cells ml$^{-1}$ by infection with 0.5–1% (v/v) of V$_1$ baculovirus supernatant. 72 h post cell proliferation arrest, insect cells were harvested by centrifugation at 800 × g for 5 min. Cell pellets were flash-frozen in liquid nitrogen and stored at −80 °C until further use.

Insect cell pellets were resuspended in 10 ml of lysis buffer comprising 50 mM HEPES-NaOH, pH 7.4, 400 mM NaCl, 10 mM imidazole, 1 mM phenylmethanesulfonyl fluoride (PMSF), 1 mM benzamidine, per 100 ml expression volume, and lysed by sonication. Lysates were cleared by centrifugation at 50,000 × g for 40 min in a Type 45 Ti fixed-angle rotor (Beckman Coulter). Pre-equilibrated Ni$^{2+}$- nitrilotriacetic acid (NTA) agarose resin (Thermo Fisher Scientific) was added to the soluble fraction and incubated for 45 min at 4 °C under agitation. Agarose resin was recovered by centrifugation and washed extensively in lysis buffer without protease inhibitors. Bound proteins were eluted in 50 mM HEPES-NaOH, pH 7.4, 400 mM NaCl, 250 mM imidazole. Eluates of immobilized metal ion affinity chromatography (IMAC) were diluted to 150 mM NaCl and loaded onto a HiTrap Heparin HP column (GE Healthcare). Protein complexes were eluted by a linear salt gradient from 150 mM to 2 M NaCl. TSEN complexes were subjected to TEV protease cleavage (1:50 protease to protein mass ratio) at 4 °C to remove the His-tag, concentrated by ultrafiltration using Amicon Ultra centrifugal filters (Merck) with a molecular weight cut-off (MWCO) of 30 kDa and polished by size exclusion chromatography on a Superdex 200 Increase 10/300 GL column (GE Healthcare) equilibrated in 50 mM HEPES-NaOH, pH 7.4, 400 mM NaCl. Peak fractions were pooled, concentrated by ultrafiltration, and flash-frozen in liquid nitrogen after supplementation with 15% (v/v) glycerol.

TSEN15–34 was typically purified from 1.6 liters of infected Sf21 suspension culture as stated above but leaving out the heparin chromatography step. IMAC eluates were buffer exchanged into 25 mM HEPES-NaOH, pH 7.4, 400 mM NaCl on a PD-10 desalting column (GE Healthcare), supplemented with TEV protease (1:50 protein to protease mass ratio), concentrated by ultrafiltration using Amicon Ultra (10 kDa MWCO) centrifugal filters (Merck) and polished on a Superdex 200 Increase 10/300 GL column (GE Healthcare) in 25 mM HEPES-NaOH, pH 7.4, 500 mM NaCl. Peak fractions were pooled, concentrated at room temperature to 25 mg ml$^{-1}$ by ultrafiltration, and diluted to 250 mM NaCl and a final protein concentration of 12 mg ml$^{-1}$ for crystallization trials.

For two-color pre-tRNA cleavage assays, viral bacmids encoding wt TSEN-STREP, wt TSEN/CLP1-FLAG, TSEN/CLP1-FLAG (TSEN2$^{H377A}$), and TSEN/CLP1-FLAG (TSEN34$^{H255A}$) pBIG2ab constructs were generated using the Tn7 transposition system in DH10EMBacY cells. The resulting bacmids were transfected into Sf9 insect cells using cellfectin II (Gibco). The virus was harvested after 3 days and used to further amplify the viral concentration in larger Sf9 cell culture. Following amplification, protein complexes were expressed in High Five cells for 72 h at 28 °C and 130 rpm which were subsequently harvested by centrifugation at 1000 × g. Cell pellets were resuspended in a purification buffer comprising 20 mM HEPES, pH 8.0, 150 mM NaCl, 1 mM MgCl$_2$, and lysed using multiple passes through a dounce homogenizer followed by sonication. The lysate was cleared via centrifugation at 28,000 × g for 40 min at 4 °C followed by filtration through a 0.45 µm filter. Purification of TSEN/CLP1-FLAG, TSEN/CLP1-FLAG (TSEN2$^{H377A}$), and TSEN/CLP1-FLAG (TSEN34$^{H255A}$) constructs was carried out via FLAG purification, using the FLAG tag carried by the CLP1 subunit. The lysate was incubated with anti-DYKDDDDK G1 affinity beads (Genscript) for 3 h at 4 °C and washed with 20 column volumes of purification buffer. The recombinant protein was eluted using 20 column columns purification buffer supplemented with 1 µM DYKDDDDK FLAG peptide (Genscript). Affinity purification of the TSEN-STREP construct was carried out using the STREP tag carried by the TSEN2 subunit. Cleared lysate was loaded onto a StrepTrap HP column (GE Healthcare) pre-equilibrated with purification buffer. Protein was eluted using a purification buffer supplemented with 5 mM D-desthiobiotin (Sigma). Following affinity purification, protein-containing fractions were pooled and loaded onto a HiTrap Q column. Protein complexes were eluted in a linear gradient from 150 mM to 2 M NaCl in 20 mM HEPES, pH 8.0, 1 mM MgCl$_2$. TSEN-containing fractions were pooled and loaded onto a Superose 6 Increase 10/300 GL column (GE Healthcare) pre-equilibrated in purification buffer. Purified TSEN complexes were analyzed by SDS-PAGE and western blotting.

For overproduction of heterotetrameric TSEN-SBP and TSEN-SBP (TSEN15$^{H116Y}$), adherent human embryonic kidney (HEK) 293 T cells were transfected with the expression plasmids with branched polyethylenimine (PEI, Sigma-Aldrich). In detail, $4 \times 10^6$ HEK293T cells were seeded the day before transfection in 100 mm dishes in DMEM medium (Gibco Life Technologies) with 10% fetal bovine serum (FBS, Capricorn Scientific) and incubated at 37 °C, 5% CO$_2$, and 90% humidity. After 24 h, cells were transfected with 13 µg of DNA and a 1:4 ratio of PEI per 100 mm dish. Transfected cells were further incubated for 48 h, detached by addition of Trypsin-EDTA (Sigma-Aldrich), and harvested by centrifugation at 500 × g for 5 min. The cell pellets were flash-frozen in liquid nitrogen and stored at −80 °C until further use. Frozen cell pellets were thawed and resuspended in 1 ml of lysis buffer containing 50 mM HEPES-NaOH, pH 7.4, 400 mM NaCl, 0.5 mM PMSF, 1.25 mM benzamidine, per 100 mm dish and lysed by sonication. Lysates were cleared by centrifugation at 20,817 × g for 1 h. Pre-equilibrated High Capacity Streptavidin agarose resin (Pierce) was added to the soluble fraction and incubated for 1 h at 4 °C under agitation. Agarose resin was

recovered by centrifugation and washed extensively in lysis buffer without protease inhibitors. Bound proteins were eluted in 50 mM HEPES-NaOH, pH 7.4, 400 mM NaCl, 2.5 mM biotin. TSEN complex eluates were subjected to TEV protease cleavage (1:20 protease to protein mass ratio) at 4 °C to remove the SBP-tag and polished by size exclusion chromatography on a Superdex 200 Increase 3.2/300 column (GE Healthcare) equilibrated in 50 mM HEPES-NaOH, pH 7.4, 400 mM NaCl. Peak fractions were pooled and subjected to pre-tRNA cleavage assays and differential scanning fluorimetry.

**Native mass spectrometry.** The buffer of purified TSEN complexes (50 µl at 1.09 mg ml$^{-1}$ for wt TSEN and 1.88 mg ml$^{-1}$ for wt TSEN/CLP1) was exchanged for 200 mM ammonium acetate buffer, pH 7.5, using 30 kDa MWCO centrifugal filter devices (Vivaspin, Sartorius). Native MS experiments were performed on a Quadrupole Time-of-flight (Q-ToF) Ultima mass spectrometer modified for transmission of high mass complexes (Waters, Manchester, UK)[70]. For data acquisition, 3–4 µl of the sample were loaded into gold-coated capillary needles prepared in-house[71]. Mass spectrometric conditions were capillary voltage, 1.7 kV; cone voltage, 80 V; RF lens voltage, 80 V; collision energy, 20 V; Aperture3, 13.6. Mass spectra were processed using MassLynx 4.1. At least 100 scans were combined. Acquired mass spectra were calibrated externally using 100 mg ml$^{-1}$ cesium iodide solution. Mass spectra were processed in MassLynx and analyzed using Massign[72].

**Phosphoprotein analysis.** To analyze the phosphorylation state of TSEN subunits, 50 µg of purified protein complexes were treated with 2000 U of Lambda Protein Phosphatase (New England Biolabs) in 200 µl dephosphorylation buffer (50 mM HEPES-NaOH, pH 7.4, 400 mM NaCl, 1 mM DTT, 1 mM MnCl$_2$) for 2 h at 30 °C. Untreated and dephosphorylated complexes were analyzed via SDS-PAGE. Gels were stained with ProQ Diamond Phosphoprotein Gel Stain (Thermo Fisher Scientific) according to the manufacturer's instructions and imaged on a Typhoon Bioimager (GE Healthcare) at excitation and emission wavelengths of 532 and 560 nm, respectively. Imaged gels were subsequently stained with InstantBlue Coomassie (Expedeon).

**Nuclear extracts.** To assay pre-tRNA splicing using patient fibroblasts, we prepared nuclear extracts. Cells from at least four confluent 15 cm dishes were tryp-sinized, the cell pellet was washed once with PBS and spun for 2 min at 300 × g. The pellet was resuspended in 1 ml 1 × PBS and transferred to a 1.5 ml tube. The tubes were centrifuged for 5 min at 137 × g. The pellet was resuspended in one volume Buffer A (10 mM HEPES-KOH pH 8.0, 1 mM MgCl$_2$, 10 mM KCl, 1 mM DTT) and incubated for 15 min on ice. A 1-ml syringe (fitted with a 0.5 mm × 16 mm needle) was filled with Buffer A and thereafter fully displaced by the plunger to remove all the remaining air within the syringe. Cells were lysed by slowly drawing the suspension into the syringe followed by rapidly ejecting against the tube wall. This step was repeated five times for complete lysis to occur. The sample was then spun for 20 s at 16,200 × g. The pellet was resuspended in two-thirds of one packed cell volume in Buffer C (20 mM HEPES-KOH, pH 8.0, 1.5 mM MgCl$_2$, 25 % (v/v) glycerol, 420 mM NaCl, 0.2 mM EDTA, 0.1 mM PMSF, 1 mM DTT) and incubated on ice with stirring for 30 min. The suspension was centrifuged for 5 min at 16,200 × g. The supernatant (corresponding to nuclear extracts) was dialyzed for 1 h against 30 mM HEPES-KOH, pH 7.4, 100 mM KCl, 5 mM MgCl$_2$, 10% (v/v) glycerol, 1 mM DTT, 0.1 mM AEBSF using dialysis membranes (Millipore 'V' series membrane). Afterwards, protein concentrations were determined (BioRad Bradford reagent), normalized using dialysis buffer, and immediately used for enzymatic assays or snap-frozen and stored at −80 °C.

**Pre-tRNA cleavage assays.** For non-radioactive assays, pUC19 vectors encoding S.c. pre-tRNA$^{Phe}_{GAA}$2-2, human pre-tRNA$^{Tyr}_{GTA}$8-1, S.c. tRNA$^{Phe}_{GAA}$2-2 and human tRNA$^{Tyr}_{GTA}$8-1 were linearized using BstNI and template DNA was isolated by agarose gel electrophoresis. RNA substrates were produced by run-off in vitro transcription using T7 RNA polymerase (New England Biolabs) and purified via anion exchange chromatography as described before with slight modifications[37,73]. Briefly, 1 µg ml$^{-1}$ of template DNA was mixed with 1000 U ml$^{-1}$ of T7 polymerase and 1.5 mM of each rNTP (New England Biolabs) in 40 mM Tris-HCl, pH 7.9, 9 mM MgCl$_2$, 2 mM spermidine, 1 mM DTT, and incubated for 4 h at 37 °C. To isolate transcribed RNAs, the reaction mixture was diluted in a 1:1 ratio (v/v) with AEX buffer containing 50 mM sodium phosphate, pH 6.5, 0.2 mM EDTA, and loaded onto a HiTrap DEAE FF column (GE Healthcare) equilibrated in AEX buffer and eluted by a linear gradient from 0–700 mM NaCl. RNA containing fractions were analyzed via denaturing urea-PAGE with subsequent toluidine blue staining. RNAs were concentrated by ultrafiltration using Amicon Ultra 3 MWCO centrifugal filters (Merck) and stored at −20 °C. 1 µg TSEN complexes were mixed with the respective RNA in a 1:5 molar ratio in 50 mM HEPES-NaOH, pH 7.4, 100 mM NaCl, 2 mM MgCl$_2$, 1 mM DTT in a 20 µl reaction volume and incubated at 37 °C for 45 min. Reactions were stopped by the addition of a 2× RNA loading buffer (95% formamide, 0.02% SDS, 1 mM EDTA) and incubation at 70 °C for 10 min. Reaction products were separated by denaturing urea-PAGE and visualized by toluidine blue staining.

For pre-tRNA cleavage assays using radioactive probes, S.c. pre-tRNA$^{Phe}_{GAA}$ (plasmid kindly provided by C. Trotta) was transcribed in vitro using the T7

MEGAshortscript kit (Ambion) including 1.5 MBq [α$^{32}$P]-guanosine-5′-triphosphate (111 TBq/mmol, Hartmann Analytic) per reaction. The pre-tRNA was resolved in a 10% denaturing polyacrylamide gel, visualized by autoradiography, and passively eluted from gel slices overnight in 0.3 M NaCl. RNA was precipitated by the addition of three volumes of ethanol and dissolved at 0.1 µM in buffer containing 30 mM HEPES-KOH, pH 7.3, 2 mM MgCl$_2$, 100 mM KCl. To assess pre-tRNA splicing, one volume of 0.1 µM body labeled S. cerevisiae pre-tRNA$^{Phe}$, pre-heated at 95 °C for 60 sec and incubated for 20 min at room temperature, was mixed with four volumes of reaction buffer (100 mM KCl, 5.75 mM MgCl$_2$, 2.5 mM DTT, 5 mM ATP, 6.1 mM Spermidine-HCl, pH 8.0 (Sigma), 100 U ml$^{-1}$ RNasin RNase inhibitor (Promega)). Equal volumes of this reaction mixture and cell extracts with a total protein concentration of 6 mg ml$^{-1}$ were mixed and incubated at 30 °C. At given time points, 5 µl of the mix were deproteinized with proteinase K, followed by phenol/chloroform extraction and ethanol precipitation. Reaction products were separated on a 10% denaturing urea-polyacrylamide gel, and tRNA exon formation was monitored by phosphorimaging. Quantification of band intensities was performed using ImageQuant software.

For two-colored pre-tRNA cleavage assays, 5′-cyanine5 (Cy5) and 3′-Fluorescin (FITC) labeled human pre-tRNA$^{Tyr}_{GTA}$ 3-1 was purchased from Dharmacon. Pre-tRNA was resuspended in nuclease-free water (New England Biolabs) at 100 µM stock concentration. Prior to use, the pre-tRNA stock was diluted 1 in 2 into RNA loading buffer (New England Biolabs) and separated on a 10% acrylamide urea-TBE denaturing gel, with the band corresponding to pre-tRNA excised. The excised bands were crushed using a pipette tip in a 1.5 ml Eppendorf tube and incubated in 300 µl of 20 mM Tris-HCl, pH 8.0, 250 mM KCl, overnight at room temperature. Gel fragments were removed by centrifugation at 17,000 × g. The supernatant was transferred to a fresh Eppendorf tube and tRNA precipitated through the addition of 4 µl RNA-grade glycogen (Thermo Fisher Scientific) and 1 ml of 100% isopropanol. The precipitate was collected through centrifugation at 17,000 × g and the pellet was washed in 75% ethanol. The resulting pellet was resuspended in nuclease-free water (New England Biolabs) and RNA was quantified through measurement of A$_{260}$ prior to storage at −80 °C. Purified pre-tRNA was diluted 1:10 in cleavage buffer (20 mM HEPES, pH 8.0, 100 mM KCl, 2.5 mM Dithiothreitol, 5 mM Spermidine-HCl, 5 mM MgCl$_2$). Pre-tRNA was incubated at 90 °C for 1 min and cooled to room temperature for 20 min to ensure folding. 20 pmols of folded pre-tRNA substrate were incubated with a final concentration of 5 U ml$^{-1}$ RNasin plus inhibitor (Promega), 5 mM ATP, and 8 pmol of TSEN complex in a final reaction volume of 20 µl for 1 h at 30 °C. RNA was extracted through the addition of 150 µl of cleavage buffer followed by 150 µl of 25:24:1 phenol:chloroform:isoamyl alcohol solution (Thermo Fisher Scientific). Samples were mixed and centrifuged at 17,000 × g for the separation of RNA and protein layers. The top layer was transferred to a fresh Eppendorf tube and RNA precipitated through the addition of 4 µl RNA-grade glycogen (Thermo Fisher Scientific) and 1 ml of 100% isopropanol. Precipitated RNA was centrifuged at 17,000 × g and the pellet was washed in 75% ethanol solution. RNA was resuspended in 10 µl of nuclease-free water (New England Biolabs). 5 µl of RNA solution was suspended in 5 µl of RNA loading buffer (95% (v/v) formamide, 10 mM EDTA). Samples were boiled at 95 °C for 10 min prior to loading on a 10% acrylamide urea-TBE denaturing gel. Results were visualized using a Typhoon FLA 9000 (GE Healthcare).

**Fluorescent 3′ end labeling of RNA.** RNAs were labeled site-specifically at their 3′ ends using periodate chemistry and a hydrazide derivate of cyanine5 (Cy5) fluorophore (Lumiprobe) as described previously[74]. Typically, 5 µM of RNA were mixed with 2.5 µl of 400 mM NaIO$_4$, 13.3 µl of 3 M KOAc pH 5.2, in a total volume of 400 µl and incubated for 50 min on ice to oxidize the 2′−3′ diols of the terminal ribose to aldehydes. Oxidized RNAs were ethanol precipitated and resuspended in 400 µl of diethylpyrocarbonate (DEPC)-treated water containing 1 mM of Cy5-hydrazide and 13.3 µl of 3 M KOAc, pH 5.2. After incubation at 4 °C overnight in the dark under agitation, RNA was ethanol precipitated and buffer exchanged to fresh DEPC-treated water using a Zeba Spin desalting column (Thermo Fisher Scientific) to remove the unreacted dye. The optical density at wavelengths of 260 and 650 nm was measured using a NanoDrop 1000 spectro-photometer (Thermo Fisher Scientific) to determine the frequency of incorporation (FOI; the number of incorporated fluorophores per 1000 nucleotides) and labeling efficiency.

**tRNA pull-down assays.** 25 µl of the monoclonal α-His antibody (Catalog-ID H1029, Sigma-Aldrich) were mixed with 100 µl buffer comprising 50 mM HEPES-NaOH, pH 7.4, 400 mM NaCl (HS buffer) and coupled to 25 µl of Protein G Agarose (Thermo Fischer Scientific) for 30 min at 4 °C under agitation. Beads were washed twice with 1 ml HS buffer (1500 × g, 3 min) and incubated with 8 µg of inactive His-tagged tetrameric TSEN complex (His$_6$-tag on TSEN15) in a total volume of 100 µl for 1 h at 4 °C. After washing three times with 150 µl buffer comprising 50 mM HEPES-NaOH, pH 7.4, 100 mM NaCl (LS buffer), 100 ng of Cy5-labeled RNA were added to the beads and incubated for 1 h at 4 °C under agitation. After binding, beads were washed again 3× in 150 µl LS buffer and bound macromolecules were eluted by addition of 5 µl 4× SDS loading buffer plus 20 µl LS buffer and incubation at 70 °C for 3 min. Eluted components were separated by

SDS-PAGE and visualized by in-gel fluorescence on an ImageQuant LAS 4000 system and immunoblotting. As positive and negative controls, the pull-down assay was performed without the addition of inactive tetrameric TSEN to the antibody-coupled beads or in the presence of an excessive amount (2 µg) of unlabeled RNA, respectively.

**Electrophoretic mobility shift assays**. 3′-Cy5-labeled pre-tRNA substrates (10 nM final) were mixed with increasing amounts of inactive tetrameric TSEN complexes (typically 10 nM up to 1 µM) in a total volume of 20 µl EMSA buffer comprising 50 mM HEPES-NaOH, pH 7.4, 100 mM NaCl, 1 mM DTT, 4% (v/v) glycerol in DEPC-treated water. After incubation for 30 min on ice in the dark, samples were loaded onto a 4% Tris-Borate-EDTA native polyacrylamide gel, which had been pre-run for 15 min at 180 V in 0.5× TBE buffer. Free and complex RNAs were separated for 1 h at 180 V at 4 °C in the dark. In-gel fluorescence was detected on an ImageQuant LAS4000 or Typhoon 9400 device (GE Healthcare) to visualize labeled RNA.

**Fluorescence anisotropy measurements**. Fluorescence anisotropy measurements were conducted on a Fluorolog-3 spectrofluorometer (Horiba) equipped with automated polarization filters at a controlled temperature of 22 °C. 120 µl of Cy5-labeled RNA with a concentration of 70 nM in 50 mM HEPES-NaOH, pH 7.4, 100 mM NaCl were titrated with TSEN complexes (1.5 µM stock) in a micro fluorescence cuvette. To avoid dilution effects, the titrant solution contained identical concentrations of the labeled RNAs. After each titration step, the solution was mixed carefully and fluorescence anisotropy was continuously assessed in 15 s increments over a period of 450 s. Anisotropy values of each data point were averaged, plotted in dependency of the protein concentration, and dissociation constants ($K_D$) were obtained by non-linear curve fitting according to a quadratic equation in Prism 5 (GraphPad Software) to compensate for non-negligible receptor concentrations. Experiments were performed in at least biological duplicates.

**Differential scanning fluorimetry**. TSEN complexes were mixed to a final concentration of 1 or 3 µM with 4x SYPRO Orange (Merck) stock in 50 mM HEPES-NaOH, pH 7.4, 400 mM NaCl. Protein unfolding was assessed on a PikoReal96 thermocycler (Thermo Fisher) by measuring SYPRO Orange fluorescence over a temperature gradient from 20–95 °C (temperature increment 0.2 °C, hold time 10 s) in a 96-well plate format. Values of technical triplicates were averaged, blank corrected, and apparent unfolding temperatures were determined as the half maximum of a sigmoidal Boltzmann fit in Prism 8 (GraphPad Software). Unfolding temperatures of PCH mutants were compared to wt TSEN complex in technical triplicates to assess their impact on stability and are representative of biological duplicates.

**Size exclusion chromatography multi-angle light scattering**. Multi-angle light scattering coupled with size exclusion chromatography (SEC-MALS) was done using a Superdex 200 Increase 10/300 GL column (GE Healthcare) at a flow rate of 0.5 ml min⁻¹ on an HPLC system composed of PU-2080 pumps, PU-2075 UV detector and degaser (JASCO) connected to a 3-angle miniDAWN TREOS light scattering detector (Wyatt Technology Corporation) and an Optilab T-rEX refractive index detector (Wyatt Technology Corporation). A BSA sample (400 µg) for calibration and 330 µg of TSEN15–34 complex at a concentration of 1.65 mg ml⁻¹ were run on a pre-equilibrated column in 25 mM HEPES-NaOH, pH 7.5, 250 mM NaCl filtered through a 0.1 µm pore size VVLP filter (Millipore). The refractive index increment ($dn/dc$) of the TSEN15–34 complex was predicted to be 0.188 ml g⁻¹ using its amino acid composition[75]. The extinction coefficient of the TSEN15–34 complex at 280 nm was calculated using the ProtParam server (https://web.expasy.org). Data analysis was accomplished using the ASTRA software package (Wyatt Technology Corporation) across individual peaks using Zimm's model for data fitting[76].

**Limited proteolysis**. Purified, full-length TSEN15–34 complex (0.9 mg ml⁻¹) was incubated with trypsin (15 µg ml⁻¹) in 50 mM HEPES-NaOH, pH 7.4, 400 mM NaCl for 1 h at room temperature. The reaction was stopped by the addition of 1 mM PMSF and the proteolyzed complex was applied to a Superdex 200 Increase 10/300 GL column (GE Healthcare) equilibrated in 50 mM HEPES-NaOH, pH 7.4, 400 mM NaCl. Peak fractions were run out on denaturing 11% SDS-PAGE and visualized by staining with InstantBlue Coomassie (Expedeon).

**Denaturating mass spectrometry**. The buffer of TSEN15–34 complexes (10 µl at 1.05 mg ml⁻¹ in 10 mM HEPES, pH 7.4, 400 mM NaCl, 0.3× Protease Inhibitor) derived from limited proteolysis was exchanged for 200 mM ammonium acetate, pH 7.5, using 3 kDa MWCO Amicon centrifugal filters (Merck Millipore). For protein denaturation, isopropanol was added to a final concentration of 1% (v/v). Subsequently, the sample was analyzed by direct infusion on a Q Exactive Plus Hybrid Quadrupole-Orbitrap mass spectrometer (Thermo Fisher Scientific) equipped with a Nanospray Flex ion source (Thermo Fisher Scientific). For this, 2–3 µl were loaded into gold-coated capillary needles prepared in-house. Spectra were recorded in positive ion mode using the following settings: capillary voltage,

2 kV; capillary temperature, 250 °C; resolution, 70.000; S-lens RF level, 50; max injection time, 50 ms; automated gain control, 1 × 10⁶; MS scan range 1000–6000 $m/z$. Approximately 300 scans were combined, and the peaks were assigned manually.

**Identification of proteins and protein fragments**. Gel electrophoresis was performed using 4–12% NuPAGE Bis-Tris gels according to the manufacturer's protocols (NuPAGE system, Thermo Fisher Scientific). Protein gel bands were excised, and the proteins were hydrolyzed as described previously[77]. Briefly, proteins were reduced with 10 mM dithiothreitol, alkylated with 55 mM iodoacetamide, and hydrolyzed with Trypsin (Roche). Extracted peptides were dissolved in 2% (v/v) acetonitrile, 0.1% (v/v) formic acid, and separated using a DionexUltiMate 3000 RSLCnano System (Thermo Fisher Scientific). For this, the peptides were first loaded onto a reversed-phase C18 pre-column (µ-Precolumn C18 PepMap 100, C18, 300 µm I.D., particle size 5 µm pore size; Thermo Fisher Scientific). 0.1% formic acid (v/v) was used as mobile phase A and 80% (v/v) acetonitrile, 0.1% (v/v) formic acid, as mobile phase B. The peptides were then separated on a reversed-phase C18 analytical column (HPLC column Acclaim® PepMap 100, 75 µm I.D., 50 cm, 3 µm pore size; Thermo Fisher Scientific) with a gradient of 4–90% B over 70 min at a flow rate of 300 nl min⁻¹. Peptides were directly eluted into a Q Exactive Plus Hybrid Quadrupole-Orbitrap mass spectrometer (Thermo Fisher Scientific). Data acquisition was performed in data-dependent and positive ion modes. Mass spectrometric conditions were: capillary voltage, 2.8 kV; capillary temperature, 275 °C; normalized collision energy, 30%; MS scan range in the Orbitrap, $m/z$ 350–1600; MS resolution, 70,000; automatic gain control (AGC) target, 3e6. The 20 most intense peaks were selected for fragmentation in the HCD cell at an AGC target of 1e5. MS/MS resolution, 17,500. Previously selected ions were dynamically excluded for 30 s and singly charged ions and ions with unrecognized charge states were also excluded. Internal calibration of the Orbitrap was performed using the lock mass $m/z$ 445.12002578. Obtained raw data were converted to .mgf files and were searched against the SwissProt database using the Mascot search engine 2.5.1 (Matrix Science).

**Crystallization, structure determination, and validation of a minimal TSEN15–34 complex**. Crystals of truncated TSEN15–34 complex (TSEN15 residues 23–170 and TSEN34 residues 208–310) were refined manually at 18 °C by mixing equal volumes of protein solution containing 12–15 mg ml⁻¹ TSEN15–34 in 25 mM HEPES-NaOH, pH 7.4, 250 mM NaCl, and crystallization solution containing 0.1 M Imidazole/MES, pH 6.5, 20% PEG3350, and 0.2 M MgCl₂ in a vapor diffusion setup. Crystals were cryoprotected by adding 20% (v/v) glycerol to the reservoir solution and flash-frozen in liquid nitrogen. Diffraction data were collected at 100 K to a resolution of 2.1 Å on beamline P14 of the Deutsches Elektronen-Synchrotron (DESY) and were processed and scaled using the X-ray Detector Software (XDS) package[79]. Crystals belong to the monoclinic space group P2₁ with two complexes in the asymmetric unit. The structure of TSEN15–34 was solved by molecular replacement with Phaser[80] within the Phenix software package[81] using a truncated poly-Ala model of the *Aeropyrum pernix* endonuclease (residues 83–169 of the I chain and residues 93–168 of the J chain) (PDB 3P1Z)[82] as a search model. The structures of the two domain-swapped TSEN15–34 dimers were manually built with Coot[83] and refined with Phenix[84] with good stereochemistry. The statistical quality of the final model was assessed using the program Molprobity[85]. Structure figures were prepared using PyMOL.

**Cell culture**. Human fibroblasts were cultured at 37 °C, 5% CO₂ in Dulbecco's modified Eagle's medium (Sigma) supplemented with 10% fetal bovine serum (Gibco), 100 U ml⁻¹ penicillin, and 100 µg ml⁻¹ streptomycin sulfate (Lonza). Cells were split and/or harvested at 80–90% confluency using 0.05% Trypsin–EDTA.

**Northern blotting**. Isolation of total RNA from cell lines was performed using the Trizol Reagent (Invitrogen) according to the manufacturer's instructions. Typically, 4–5 µg of RNA was separated in a 10% denaturing urea-polyacrylamide gel (20 × 25 cm; Sequagel, National Diagnostics). The RNA was blotted on Hybond-N + membranes (GE Healthcare) and fixed by ultraviolet cross-linking. Membranes were pre-hybridized in 5× SSC, 20 mM Na₂HPO₄, pH 7.2, 7% SDS, and 0.1 mg ml⁻¹ sonicated salmon sperm DNA (Stratagene) for 1 h at 80 °C (for DNA/LNA probes) or 50 °C (for DNA probes). Hybridization was performed in the same buffer overnight at 80 °C (for DNA/LNA probes) or 50 °C (for DNA probes) including 100 pmol of the following [5′-³²P]-labeled DNA/LNA probe (Exiqon, Denmark; LNA nucleotides are indicated by "*X"): tRNA^Ile_TAT1-1 5′ exon probe, 5′-TA*T AA*G TA*C CG*C GC*G CT*A AC-3′, or the following DNA probe: tRNA^Ile_TAT1-1 intron probe, 5′-TGC TCC GCT CGC ACT GTC A-3′. Blots were subsequently washed twice at 50 °C with 5× SSC, 5% SDS and once with 1× SSC, 1% SDS and analyzed by phosphorimaging. Membranes were re-hybridized at 50 °C using a DNA probe (5′-GCA GGG GCC ATG CTA ATC TTC TCT GTA TCG-3′) complementary to U6 snRNA to check for equal loading. Quantification of band intensities was performed using ImageQuant software.

**Antibodies**. Rabbit polyclonal antibodies raised against TSEN2 (N-NGDSGKSGGVGDPREPLG-C), TSEN34 (N-QASGEQEEAGPSSSQAGPSNG-C), and TSEN54 (N-RSRSQKLPQRSHGPKDFLPD-C)[23] (Gramsch Laboratories, Schwabhausen, Germany) were affinity-purified from rabbit sera and eluted sequentially by addition of 1.5 M $MgCl_2$ and 0.1 M glycine, pH 2.5. Eluates were dialyzed against HEPES-buffered saline (HBS, pH 7.9) overnight, supplemented with 10% (v/v) glycerol, and stored at −80 °C. Small-scale pilot experiments were set up to assess the suitability of the affinity-purified antibodies for immunoprecipitation experiments using total cell lysate from HEK 293 cells. Antibodies used in western blotting (WB) and immunoprecipitation (IP) were: anti-TSEN15 (rabbit polyclonal, Atlas Antibodies, HPA029237; WB dilution 1:1000), anti-TSEN34 (IP, WB dilution 1:5000), anti-TSEN54 (WB dilution 1:5000), anti-TSEN2 (IP, WB dilution 1:5000), anti-GAPDH (rabbit monoclonal, 14C10, Cell Signaling Technology, #2118; WB dilution 1:1000), anti-β-actin (mouse monoclonal, clone AC-74, Sigma-Aldrich, A2228; WB dilution 1:3000), anti-mouse IgG–peroxidase conjugate (secondary goat polyclonal, Sigma-Aldrich, A2554; WB dilution 1:20.000), anti-rabbit IgG–peroxidase conjugate (secondary goat polyclonal, Sigma-Aldrich, AP307P; WB dilution 1:20.000), anti-polyHistidine (mouse monoclonal, clone HIS-1, Sigma-Aldrich, H1029; IP).

**Immunoprecipitation of TSEN components**. Affinity-purified antibodies against TSEN2, TSEN34, and TSEN54[36] were cross-linked to agarose beads, as described[86]. Briefly, bead-bound antibodies were incubated in 20 mM dimethylphenol (DMP), 200 mM sodium tetraborate at RT, and the reaction was then stopped by transferring the beads to 200 mM Tris-HCl, pH 8.0. After washing 3× with TBS/0.04% Triton-X-100, beads were stored at 4 °C. For immunoprecipitation (IP), total cell lysates were prepared from fresh or frozen cell pellets of non-immortalized fibroblasts as described[87]. Upon centrifugation at 16,000 × g, clear lysates were collected, protein concentration was measured, and equal amounts of total protein for each sample were used for the IPs. Upon incubation with cell lysates for 90 min at 4 °C while rotating in 1.5 ml tubes, TSEN complex-bound beads were washed 3× in 20 mM Tris-HCl, pH 7.5, 150 mM NaCl, 10% (v/v) glycerol, 0.1% (v/v) NP40, 2 mM $MgCl_2$ and split into two aliquots; one was used for a pre-tRNA splicing assay and the other was boiled in SDS-PAGE loading buffer. Pre-tRNA splicing assay was performed as described above, omitting the proteinase K treatment and the phenol/chloroform extraction and ethanol precipitation steps. Instead, aliquots were collected at indicated time-points in tubes already containing an equal amount of 2 × loading buffer and stored at −20 °C. Protein samples were analyzed by SDS-PAGE and immunoblotting.

**Hydro-tRNA sequencing**. tRNA sequencing was performed using the hydro-tRNAseq protocol, as described previously[4]. Briefly, total RNA from human-derived fibroblasts was resolved on 12% urea-polyacrylamide gel, followed by recovery of the tRNA fraction within a size window of 60–100 nt. The eluted fraction was subjected to limited alkaline hydrolysis in 10 mM $Na_2CO_3$ and $NaHCO_3$ at 60 °C for 1 h. The hydrolyzed RNA was dephosphorylated and rephosphorylated to reconstitute termini amenable for sequential adapter ligation. Fragments of 19–35 nt were converted into barcoded cDNA libraries, as described previously[88], and sequenced on an Illumina HiSeq 2500 instrument. Adapters were trimmed using cutadapt (http://journal.embnet.org/index.php/embnetjournal/article/view/200/458). Sequence read alignments and analysis were performed as described previously[4]. Split read counts were used for multi-mapping tRNA reads. Precursor tRNA reads spanned the junctions between mature sequences and leaders, trailers, or introns. Analysis for Fig. 5c, and Supplementary Fig. 8a,b was conducted in R (version 4.0.4, https://www.r-project.org/), Python (version 3.7, http://www.python.org), and Perl (version 5.18.4, https://www.perl.org/). Custom scripts are available upon request. Figures were produced using the R package ggplot2 and Prism 8 (Graphpad Software).

**Sequence alignments**. Sequence alignments were done with Clustal Omega[89] and visualized using ESPript 3.0[90]. Alignments of pre-tRNAs and tRNAs were manually edited in Jalview[91].

**Statistical analysis**. Student's two-tailed unpaired $t$-tests were carried out to determine the statistical significance of differences between samples. A $P$-value less than 0.05 was considered nominally statistically significant for all tests.

**Patient recruitment and ascertainment**. Patients suspected for PCH were submitted to the pediatric neurology of the Academic Medical Centre (AMC) for diagnostics. Primary fibroblast cell lines were generated from skin biopsies taken for diagnostic procedures. As soon as DNA diagnostics became available, patient DNA was subjected to genetic analyses. DNA sequencing confirmed the diagnosis, and the mutations were confirmed in the fibroblast lines. All procedures were performed with the full consent of the legal representative and approval of the Institutional Review Board (IRB), Amsterdam UMC, The Netherlands (#W17_090# 17.098).

**Reporting summary**. Further information on research design is available in the Nature Research Reporting Summary linked to this article.

## Data availability

The data that support this study are available from the corresponding author upon reasonable request. Atomic coordinates and structure factors were deposited to the Protein Data Bank (http://www.rcsb.org) under accession number PDB ID 6Z9U. The mass spectrometry proteomics data were deposited to the ProteomeXchange Consortium via the PRIDE partner repository with the dataset identifier PXD019034. Hydro-tRNAseq data were deposited to the Gene Expression Omnibus (GEO) repository under accession code GSE151236. Source data for Figs. 1–6 and Supplementary Figs. 1–9 are provided with this paper. Source data are provided with this paper.

## Code availability

Custom scripts for R (version 4.0.4, https://www.r-project.org/), Python (version 3.7, http://www.python.org), and Perl (version 5.18.4, https://www.perl.org/) are available upon request.

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

## Acknowledgements

We thank Rupert Abele for the analysis of SEC-MALS experiments, Jan Erik Schliep for DSF data analysis with the ProteoPlex algorithm, and Imre Berger for providing the MultiBac reagents. The synchrotron MX data were collected at beamline P14 operated by EMBL Hamburg at the PETRA III storage ring (DESY, Hamburg, Germany). We thank Gleb Bourenkov for the assistance in using the beamline. S.T. acknowledges Robert Tampé and all members of his group for discussions and comments on the manuscript and excellent administrative and technical support. S.P. thanks Kristina Uzunova for sharing her expertise in antibody purification and protein biochemistry. M.B. and C.S. acknowledge funding from the Federal Ministry for Education and Research (BMBF, ZIK program, 03Z22HN22), the European Regional Development Funds (EFRE, ZS/2016/04/78115) and the MLU Halle-Wittenberg. This study was furthermore supported by grants of the German Research Foundation (grant number TR 1711/1-7) to S.T., the Austrian Science Fund (grant number FWF P29888) to J.M. and S.T., the CRC 902 Molecular Principles of RNA-based Regulation (S.S. and S.T.), and a Boehringer Ingelheim Fonds fellowship to S.S.

## Author contributions

S.S., P.D., A.P., K.H. and S.T. expressed, purified, and prepared protein complexes from insect or mammalian cells. S.S., P.D., A.P. and S.P. performed biochemical assays. E.P.R. cloned and purified FLAG- and STREP-tagged TSEN complexes and performed dual-color pre-tRNA cleavage assays. S.P. and S.W. performed pre-tRNA splicing assays and northern blots. S.P. performed IP experiments on human fibroblasts. M.B. and C.S. conducted MS experiments and analyzed the data. S.S. and S.T. performed crystallography experiments, collected X-ray diffraction data, and built the atomic model. F.B. generated cell lines of PCH patient-derived fibroblast. T.G. performed hydro-tRNAseq experiments and bioinformatic analyses under the supervision of T.T. J.M. and S.T conceived the project, supervised the work, and designed the experiments. S.S., P.D. and S.T. wrote the initial draft of the paper with input from all authors. J.M. and S.T. acquired funding.

## Funding

## Competing interests

The authors declare no competing interests.
