## [Peer Review File · Nature Communications]

Assembly defects of the human tRNA splicing endonuclease contribute to impaired pre-tRNA processing in pontocerebellar hypoplasiaReviewers' Comments:

Reviewer #1:

Remarks to the Author:

1. The work in perspective: The eukaryotic pre-tRNA splicing endonuclease was discovered more than 40 years ago and its heterotetrameric subunit composition has been established since for yeast (1997) and human cells (2004). Despite this history, the structure of this key enzyme has remained unknown because it seemed not to be possible to reconstitute the complex from recombinant subunits. This situation changed this year with the publication from the Stanley group (Hayne et al. NAR) and Sekulovski et al. (submitted to Nature Communications). The breakthrough is due to co-expression of multiple subunits. Here, Sekulovski et al. provide convincing evidence that the active heterotetramer can be assembled in vitro from co-expressed inactive heterodimers, TSEN15/TSEN34 with TSEN2/TSEN54 (Fig. 1). In contradiction to earlier models, both groups report, surprisingly, that the RNA kinase subunit, CLP, is not required for complex assembly nor enzyme activity. The studies by Sekulovski et al. regarding TSEN assembly and the role of CLP1 are well conducted; the data are completely convincing, but the authors do not do a very good job of putting their work in perspective with Hayne et al.

2. Structural analyses: A major advance in Sekulovski et al. is the structural analysis. The authors provide a 2.1 angstrom structure of a partial TSEN15/TSEN34 dimer and they document that the structure is conserved with the Archaeal enzyme (Fig. 3). They also conduct in vitro experiments to confirm that the human enzyme binds the mature domain of intron-containing pre-tRNA and that the conserved A-I base pair between the anticodon and the intron are necessary for pre-tRNA cleavage (Fig. 2). Again, the data are important and completely convincing. Perhaps the authors should consider discussing how their data might be similar/different from Hayne et al. that documented that human TSEN can employ tRNA anticodon stem loop RNAs as substrates.

3. Effects of PCH mutations upon TSEN thermos stability: Sekulovski et al. study the consequences of TSEN mutations that are causative of pontocerebellar hypoplasia (PCH). TSEN subunits with PCH mutations assemble into heterotetrameric complexes; however, as determined by differential scanning fluorimetry, the mutant complexes are less thermosensitive than wild-type complexes (Fig. 4). These results are novel and important.

4. TSEN complexes and their activities for PCH enzyme: In an effort to gain an understanding how TSEN mutations cause PCH, the authors study the TSEN complexes and their activities isolated from fibroblasts of control individuals, individuals with PCH TSEN54 heterozygous mutations, and individuals with homozygous TSEN54 PCH mutations (Fig. 5 and 6). In general, the results from these studies are often subtle and statistical significances of the data are not provided. Further, the various different experiments often employ different cell lines.

Fig. 5a. Results concerning in vitro splicing for the 4 control strains are not consistent; for example, for Ba2, unlike the other 3 cell lines, there is little mature accumulation, although tRNA halves are evident; the other 3 control samples appear to generate mature tRNA. Are the results comparing the 2 homozygous PCH cell lines to the 4 control cell lines statistically significant?

Fig. 5b. The Northern data are not quantitated; are the data statistically significant? The ratios between pre-tRNA^{Ile} to U6 and mature tRNA should be provided in this panel.

Fig. 5c. The results are variably subtle; for example, homozygous cell line, Ba18, pre-tRNA^{Ile} to U6 is 1.3, not very different than 1.0 for the control and less than the Ba12 (2.2) and Ba17 (1.5) heterozygous cell lines. It appears that this experiment has only been performed on 2 independent RNA extractions. Data from this assay do not appear to clearly demonstrate that there is a difference for pre-tRNA splicing between PCH and non-PCH cell lines. Minor: the legend reads mature tRNA^{Ile}TAT1-1, Northern analyses of mature tRNAs should not distinguish between the mature tRNAs encoded by the various loci.

Fig. 5c. It would be valuable to provide raw data for individual pre-tRNAs rather than to lump all intron-containing tRNAs together; there appears to be much scatter in the data (Extended data, Fig.

5a). Extended data, Fig. 5b: Is there an explanation for the differences in splicing of pre-tRNAs encoded by different loci? This seems to be unexpected. Are the data statistically significant?

5. Altered TSEN composition in cells with PCH mutations: Although the authors detected no differences in the *in vitro* assembly of the TSEN complex comparing WT enzyme to enzyme containing the PCH TSEN54 A307S mutation, assessment of this *in vivo* by pull-down of TSEN2 appears to result in reduced levels of co-purified PCH TSEN54 (Fig. 6b). These results are very interesting, but the data need to be quantitated for the ratio of IP vs input of TSEN54 and the IP for TSEN54 vs. nonspecific GAPDH and statistical significances should be reported. Nevertheless, the on-bead tRNA cleavage assay provides strong data in support of the hypothesis that TSEN complexes containing a TSEN54 PCH allele are defective in pre-tRNA cleavage, perhaps providing the strongest evidence that reduced pre-tRNA cleavage is causative for PCH; however, is it possible to normalize the data to levels of complete TSEN complexes?

6. Discussion:

Cause of PCH: Since PCH cells appear not to have decreased levels of mature tRNAs, is it not possible that the phenotypes result from aberrant increased levels of pre-tRNAs? Perhaps this should be added to the other possibilities invoked.

Discrimination of pre-tRNAs from mature tRNAs: The authors should elaborate upon what is meant by substrate selection resulting from different binding kinetics between pre-tRNAs and mature tRNAs. Perhaps, it would be valuable to mention that since splicing occurs in the nucleus of human cells, that the higher concentration of pre-tRNA substrate to mature tRNA in this organelle might contribute to specificity. The authors should also address the studies from Hayne et al. showing that tRNA fragments containing the anticodon stem loop + intron serve as substrates *in vitro*.

Reviewer #2:

Remarks to the Author:

"Assembly defects of the human splicing endonuclease...."by Sekulovski et al

The Sekulovski paper focuses on the human 4-protein TSEN (tRNA splicing endonuclease) complex and the question of why mutations in the proteins of that complex are associated with pontocerebellar hypoplasia---a group of hereditary neuromuscular disorders. The TSEN complex associates with the RNA kinase CLP1, and roughly 6% of several hundred genome -encoded tRNAs need this system to create specific mature tRNAs.

Here the authors do a detailed biochemical/biophysical analysis of the recombinant WT and mutant complexes, together with an investigation of overall tRNA splicing events in patient fibroblasts. The work is of high quality and presented in a long-winded somewhat tedious fashion, as if the authors are trying to figure out how it all fits together. In the end, in spite of an enormous amount of quality work, they are not able to provide convincing clarity on why mutations in the proteins of TSEN complex are associated with pontocerebellar hypoplasia. And yet, much is learned about the biochemical personality and characteristics of the TSEN heterotetramer. With some further work with patient samples, which may inherently be difficult to achieve, this study could advance understanding closer to establishing a clear causal rationale for the disease phenotype.

Positives

Confirm role in mammals of an A:I base pair for directing splice-site selection, as reported previously in *Drosophila*

Do sophisticated protein manipulations to prepare and assemble a heterotetrameric TSEN2-15-34-54

stoichiometric complex and take that tetramer with CLP1 to give a heteropentameric complex. Impressive work.

Also, deconvoluted CLP1 and ATP from the rest of system. Did this by separately assembling an inactive heterodimeric TSEN2-54 and an inactive heterodimeric TSEN15-34 complex, which when combined together give endonuclease activity in the absence of CLP1 and ATP. Nice way to show heterotetrameric TSEN complex is necessary and sufficient for cleavage activity, and that ATP and CLP1 have separate roles from catalysis per se on the endonuclease side of the overall splicing reaction.

Show that pre-tRNA substrate recognition is based on parts of pre-tRNA that are kept in the mature tRNA product, and that an intron per se is not needed for TSEN-pre-RNA complex stability.

Determined crystal structure of a stable proteolytic complex of the TSEN15-34 heterodimer. This required a laborious trial and error process to obtain a stable fragment. This dimer is missing substantial portions of the two constituent monomers of the heterodimer. They observe the well-established endonuclease fold. With further work they conclude that tRNA endonucleases for splicing have a common ancestor, which is a conclusion that contrasts with that of a prior solution NMR study of the homodimeric TSEN15 that evidently claimed just the contrary.

Construct 4 different, individual PCH-associated mutations in TSEN2, 34, and 54. In pull downs, the introduced mutations had no effect on the protein-protein interactions, subunit compositions, or precursor tRNA cleavage kinetics. They did however find that the mutant complexes had lower thermal stabilities, being diminished in their denaturation assays by 1 to 7 degrees, depending on the mutant complex.

They acquired patient fibroblasts harboring the most common PCH-associated mutation, namely, an A307S substitution in TSEN54. They establish that mutant cells have an increased level of precursor tRNAs, which of course are intron-containing. They also show that, in these cells, the TSEN assembly is defective when compared to wild-type patient fibroblasts. This defect is not due to a reduction in the levels of the respective proteins in the patient cells.

Negatives

While the work is carefully, thoughtful, and presented in well-written English, it reads as a collection of observations rather than as a cohesive story. I think the main disappointment is the lack of more convincing evidence that the PCH phenotype is caused by assembly defects in the TSEN complex, which in turn affect the output of mature tRNAs. What's missing are observations on patient fibroblasts beyond those harboring just the most common A307S substitution. Curiously, of the four recombinant mutant constructs they investigated, the A307S TSEN54 had the smallest effect on complex stability, as measured by thermal denaturation. Barely over 1 degree. And yet, the cells harboring this mutation had TSEN assembly defects. In contrast, the assembly of recombinant complexes was unaffected by this 'mild' or even by stronger mutations. One way to interpret this paradox is that, in the patients, the assembly defect and the negative effects on tRNA processing are due to factors in cells that are yet uncharacterized and that possibly context effects are of over-riding importance. For example, the state of phosphorylation of TSEN? Or the disposition of CLP1? In the last paragraph of the Discussion the authors allude broadly to ambiguities. But they seem to overlook, or at least not comment on, the specific path forward that would help them here.

Also, the authors suggest that the severity of the disease may correlate with the magnitude of the lowered denaturation temperatures of the recombinant complexes of mutant proteins. I realize the difficulty, but if patient fibroblasts encoding the other mutant proteins were examined, then a correlation of the strength of the assembly defect in-step with the degree of pre-tRNA accumulation would provide nice support for the idea that, indeed, the lowered denaturation temperatures are close

to, or at, the root cause of the overall disease phenotype.

I feel the reconstitution experiments, activity and melting curves analyses of the recombinant complexes, and the analysis of patient fibroblasts are the main story here. The x-ray structure of the TSEN15-34 complex is of less significance because so much of the complex is missing and only the truncated WT complex was examined. Also, the structure in some ways is confirmatory.

I agree that concluding the determined structure of the complex suggests it comes from a common ancestor, while contradicting an earlier conclusion by others, is useful and interesting. And yet really has no connection with the story here.

Although the Discussion is well written and covers a lot of ground, it does go on and on without a sharp cohesive focus.

I would like to see if the authors can get some additional patient fibroblasts and resubmit with a much 'tighter' story.

Reviewer #3:

Remarks to the Author:

To whom it might concern

The authors describe the recombinant production of tetrameric TSEN complex produced in human cells or in a baculovirus expression system. They biochemically analyze the interaction with substrate/product tRNAs. They map the substrate interaction to the mature domain of the target tRNA. The recombinant complex is active in intron excision. Neither 15/34 or 2/54 sub-dimers are active, which points towards a composite active site of the complex. The authors solve the crystal structure of the 15/34 dimer. A disease mutation lies in the interface of the dimer but does not disrupt dimer formation in vitro. However, this disease mutation, as well as other disease mutations tested, decrease the thermal stability of the complex. Furthermore, protein levels and pre/mature tRNA levels are compared in patient fibroblast samples. While overall protein levels remain undistinguishable, unspliced tRNA target species accumulate, which points towards intron excision defects in the patient samples. Co-IP points towards complex assembly defects in patient samples. The authors conclude that the thermal destabilization of the TSEN protein complex, paired with its low abundance leads RNA processing defects and cause some forms of pontocerebellar hypoplasia.

The study is sound, conclusive and technically comprehensive in itself. The protocol for recombinant production of the active TSEN complex is noteworthy and opens the way to further structural and mechanistical examination (p.ex. cryo-EM) of the mechanism of tRNA splicing and recognition of non-tRNA targets of the complex. In contrast to a previous study in E.coli, (Hayne et al. 2020), the authors report the existence of two stable sub-dimers in the eukaryotic expression systems. Biochemical testing of the substrate requirements has been performed by mutational analysis of the codon-anticodon interaction and comparison of pre- and mature tRNA. The main hypothesis, that thermal destabilization of the complexes leads to disease causing RNA processing defects, independent from CLP1, is convincingly cross validated by the analysis of patient samples.

Please address the following minor comments:

Line 55: "low copy number per cell" versus line 93: "High expression of TSEN54 in neurons of the pons, cerebellar dentate, and olivary nuclei" – this is contradictory, you have to relativize the context. Attention, in line 249 you cite different references than in 55.

Line 57: challenged in which respect?

Line 193: Better: The domain swab is most likely a crystal-packing artefact, since the complex shows a homogenous 1:1 TSEN15:TSEN34 stoichiometric behavior in solution, as shown by SEC-MALS

analysis.

Line 197: "C-terminal" instead of "N-terminal"

Figure 3 a, b, c, d : the figures are too small to see the side chain details. Moreover it is confusing how the different views relate to each other.

Some nice-to-have-discussed points, which I would not see as prerequisite for publication though:

Are there known human TSEN SNPs that are not malign, and would they keep the complex thermostable?

Do all the known disease mutations lie in the complex interfaces?

How conserved are TSEN 15/34 to 2/54, would you imagine a symmetric dimer of dimers? Could the domain-swab observed in the crystal structure resemble the mode of interaction between both TSEN heterodimers to form the tetramer? Extended Figure 1d would also indicate recombinant TSEN2/54 has the tendency to tetramerize.

There are other studies linking decreased protein stability to disease that you could discuss/reference to support your observation.

Please do not hesitate to contact me for further discussion.

Best regards,
Eva Kowalinski

Reviewer #1 (Remarks to the Author):

1. The work in perspective: The eukaryotic pre-tRNA splicing endonuclease was discovered more than 40 years ago and its heterotetrameric subunit composition has been established since for yeast (1997) and human cells (2004). Despite this history, the structure of this key enzyme has remained unknown because it seemed not to be possible to reconstitute the complex from recombinant subunits. This situation changed this year with the publication from the Stanley group (Hayne et al. NAR) and Sekulovski et al. (submitted to Nature Communications). The breakthrough is due to co-expression of multiple subunits. Here, Sekulovski et al. provide convincing evidence that the active heterotetramer can be assembled in vitro from co-expressed inactive heterodimers, TSEN15/TSEN34 with TSEN2/TSEN54 (Fig. 1). In contradiction to earlier models, both groups report, surprisingly, that the RNA kinase subunit, CLP, is not required for complex assembly nor enzyme activity. The studies by Sekulovski et al. regarding TSEN assembly and the role of CLP1 are well conducted; the data are completely convincing, but the authors do not do a very good job of putting their work in perspective with Hayne et al.

We thank reviewer 1 for this very positive evaluation of our work. We have put the work of Hayne and colleagues ([PMID: 32476018]; Nucleic Acids Res. 2020 Aug 20;48(14):7609-7622) in perspective with ours and have changed the text accordingly. Furthermore, we also added the recent work of Hurtig and coworkers ([PMID: 33649230]; Proc Natl Acad Sci U S A. 2021 Mar 9;118(10):e2020429118.), who showed that the yeast TSEN complex is involved in processing messenger RNAs that encode mitochondrial proteins.

Results: *'In line with data from reconstituted TSEN recombinantly produced in Escherichia coli³⁵, these observations indicate that active human TSEN assembles from non-functional, heterodimeric submodules independently of CLP1 and ATP.'*

Discussion: *'TSEN was shown to cleave artificial intron-containing anticodon stem loop structures³⁵. A three-dimensional structure of TSEN with substrate RNA will help define how eukaryotic TSEN recognizes pre-tRNAs and anticodon stem loop structures in particular^{35,42}.' ... 'TSEN and CLP1 are key factors in the generation of tRNA intronic circular (tric) RNAs, a poorly characterized class of short non-coding RNAs in Drosophila and humans^{35,46}.'*

2. Structural analyses: A major advance in Sekulovski et al. is the structural analysis. The authors provide a 2.1 angstrom structure of a partial TSEN15/TSEN34 dimer and they document that the structure is conserved with the Archaeal enzyme (Fig. 3). They also conduct in vitro experiments to confirm that the human enzyme binds the mature domain of

intron-containing pre-tRNA and that the conserved A-I base pair between the anticodon and the intron are necessary for pre-tRNA cleavage (Fig. 2). Again, the data are important and completely convincing. Perhaps the authors should consider discussing how their data might be similar/different from Hayne et al. that documented that human TSEN can employ tRNA anticodon stem loop RNAs as substrates.

We again appreciate the positive evaluation of our structural work and have discussed these findings in light of the data of Hayne and colleagues (see above).

3. Effects of PCH mutations upon TSEN thermos stability: Sekulovski et al. study the consequences of TSEN mutations that are causative of pontocerebellar hypoplasia (PCH). TSEN subunits with PCH mutations assemble into heterotetrameric complexes; however, as determined by differential scanning fluorimetry, the mutant complexes are less thermosensitive than wild-type complexes (Fig. 4). These results are novel and important.

We agree with reviewer 1 and thank him/her for emphasizing the novelty of our work.

4. TSEN complexes and their activities for PCH enzyme: In an effort to gain an understanding how TSEN mutations cause PCH, the authors study the TSEN complexes and their activities isolated from fibroblasts of control individuals, individuals with PCH TSEN54 heterozygous mutations, and individuals with homozygous TSEN54 PCH mutations (Fig. 5 and 6). In general, the results from these studies are often subtle and statistical significances of the data are not provided. Further, the various different experiments often employ different cell lines.

Fig. 5a. Results concerning in vitro splicing for the 4 control strains are not consistent; for example, for Ba2, unlike the other 3 cell lines, there is little mature accumulation, although tRNA halves are evident; the other 3 control samples appear to generate mature tRNA. Are the results comparing the 2 homozygous PCH cell lines to the 4 control cell lines statistically significant?

We appreciate the reviewer's comments referring to apparently inconsistent results of the 4 control cell lines in the in vitro pre-tRNA splicing assay (i.e. Ba2 showing little mature tRNA formation in comparison to the other 3 control cell lines). Two enzymatic activities can be monitored by the tRNA splicing assay in cell lysates: (1) intron excision by the TSEN complex, and (2) subsequent exon ligation by concerted action of the tRNA ligase complex together with its co-factor archease. Our results in Figure 5a undoubtedly reveal an impairment of TSEN cleavage activity in cell lysates of the two patient (homozygous)

TSEN54^{A307S} cell lines, in contrast to the exon generation of all 4 control cell lines at comparable efficiencies. Therefore, in our opinion, the claim of "observing a reduction in pre-tRNA splicing efficiency in homozygous TSEN54 c.919G>T cell lines compared to control cell lysates" in the current manuscript version is correct. However, the subsequent exon ligation is impaired in control cell line Ba2. This reaction step exclusively depends on the activity of the tRNA ligase complex. In our laboratory we have recently shown that the biochemical activity of complex members of the tRNA ligase complex is subject to regulation by other co-factors (e.g. archease triggering multiple-turnover reactions of the tRNA ligase) and oxidative stress ([PMID:23474986]; Nature. 2013 Mar 28;495(7442):474-80.). Thus, variations in exon ligation may depend on cellular archease levels and/or redox environment depending on the genetic background of the cell lines. In the past we have observed a similar, unequal tRNA ligation activity within a control cell line group (see Figure S2A in [PMID: 24766809]; Cell. 2014 Apr 24;157(3):636-50). If the reviewer agrees, we prefer keeping Figure 5a unchanged in the current version of the manuscript, where we have already stated in the main text that "subtle differences in ligation efficiency, as observed for cell line Ba2, may result from the fibroblasts having different genetic backgrounds."

Fig. 5b. The Northern data are not quantitated; are the data statistically significant? The ratios between pre-tRNA^{lle} to U6 and mature tRNA should be provided in this panel.

The reviewer had concerns about the lack of quantification and statistical significance of the data. We wish to point out that the sole reason for performing this specific northern blot analysis was to rule out any possible tRNA intron accumulation in PCH patient cell lines as occurs in CLP1^{R140H} fibroblasts (see [PMID: 24766809]; Cell. 2014 Apr 24;157(3):636-50). This analysis, together with the new Source Data 10d (corresponding to Fig. 5d) using a tRNA^{lle}_{TAT1-1} intron probe, revealed no apparent tRNA intron accumulation, therefore ruling out defects in downstream processes of the tRNA splicing reaction other than intron excision.

Fig. 5c. The results are variably subtle; for example, homozygous cell line, Ba18, pre-tRNA^{lle} to U6 is 1.3, not very different than 1.0 for the control and less than the Ba12 (2.2) and Ba17 (1.5) heterozygous cell lines. It appears that this experiment has only been performed on 2 independent RNA extractions. Data from this assay do not appear to clearly demonstrate that there is a difference for pre-tRNA splicing between PCH and non-PCH cell lines. Minor: the legend reads mature tRNA^{lle}TAT1-1, Northern analyses of mature tRNAs should not distinguish between the mature tRNAs encoded by the various loci.

The reviewer referred to “subtle results“ we obtained from our studies focusing on the enzymatic activity of the TSEN complex in control and patient cell lines. We agree with the reviewer in that the differences between control and patient (homozygous) cell lines are rather minor, but we would like to point out that this might be expected, given that PCH fibroblasts do not display any obvious phenotype and their growth rate is comparable to that of controls. A relatively modest accumulation of pre-tRNAs in patients is in line with the notion that the residual TSEN activity is sufficient to sustain tRNA production and growth. We have now included this comment in the revised version of the text. We agree with the reviewer’s remark that variations among different cell lines sharing the same TSEN genotype can be occasionally observed. This is however due to dealing with samples derived from unrelated individuals, and therefore not genetically identical. Despite these differences, we have identified a clear trend in patient cells. Overall, we have been able to investigate for the first time TSEN activity in PCH patient fibroblasts describing impaired cleavage efficiency and reduced stability of the complex. As PCH mutations impact only neurons *in vivo*, causing cerebellar degeneration, the use of patient-derived neuronal cell lines might be a better source of material for RNA analysis in future studies, as differences in pre-tRNA levels might be more pronounced. The generation of these cells could be pursued by e.g. iPS cell technologies coupled to a specific program of cerebellar differentiation, although established protocols for the latter are not currently available. While this strategy will certainly help future studies build a model for PCH, it is unlikely to provide an efficient tool for biochemical analysis, due to the difficulty in obtaining cell numbers comparable to the amount of fibroblasts we used in this work.

The reviewer also referred to the „lack of statistical significances of the data“, due to the low number of samples examined. We agreed with the reviewer on this point and have therefore repeated these experiments and re-evaluated our northern blot analysis. We have re-cultivated a comprehensive set of fibroblast cell lines comprising three TSEN genotype classes: 1) control (n=3); 2) heterozygous TSEN54^{A307S} (n=4); 3) patients (homozygous) TSEN54^{A307S} (n=8). We performed independent triplicate RNA isolations (replicate 1-3) for each cell line and performed northern blot analysis using an intron (DNA) probe to assess pre-tRNA^{lle}_{TAT}1-1 levels, and a 5´ exon (LNA, locked nucleic acid) probe for mature tRNA^{lle}_{TAT} detection (new Figure 5d). We quantified ratios of pre-tRNA^{lle}_{TAT}1-1 (DNA or LNA probe) to either mature tRNA^{lle}_{TAT} (LNA probe) or U6 snRNA levels (new Figure 5e). As already mentioned, despite some variations within the same TSEN genotype class (e.g. cell line Ba10) and replicates of the same cell line, statistically significant differences between groups of control and patient cell lines were obtained using an unpaired Student’s t-test for ratios of pre-tRNA^{lle}_{TAT}1-1 (DNA probe) to mature tRNA^{lle}_{TAT} (LNA probe; 1.2-fold; p-value=0.0371) or

U6 (1.3-fold; p -value=0.0344). We now provide these data in the revised version as Figure 5d (representative Northern blot and quantification of triplicate samples) and Figure 5e (statistical group analysis). Accordingly, we have also revised the Source Data figure 10d. We provide an overview of the triplicates with statistical analysis for the reviewer below [Fig. R1.a]. We wish to emphasize that differences in pre-tRNA levels measured by northern blot analysis, albeit subtle, were independently confirmed by the more sensitive and robust hydro-tRNAseq technology, which revealed an accumulation of ~2-6 fold of intron-containing pre-tRNAs on an RNome-wide scale.

Minor comment of the reviewer: figure legend is corrected in the new version (mature $tRNA^{Ile}_{TAT1-1}$ changed to mature $tRNA^{Ile}_{TAT}$).

Figure R1.a

Fig. R1.a – Northern blot analyses comparing pre-tRNA^{Ile}_{TAT} 1-1 levels to levels of mature tRNA^{Ile}_{TAT} or U6 snRNA in control fibroblasts and fibroblasts carrying the heterozygous or

homozygous TSEN54 c.919G>T mutation. Shown are three independent experiments. Signal intensities were quantified and displayed as ratios normalized to Ba13 in the bottom panel.

Fig. 5c. It would be valuable to provide raw data for individual pre-tRNAs rather than to lump all intron-containing tRNAs together; there appears to be much scatter in the data (Extended data, Fig. 5a). Extended data, Fig. 5b: Is there an explanation for the differences in splicing of pre-tRNAs encoded by different loci? This seems to be unexpected. Are the data statistically significant?

Response to "Fig. 5c. It would be valuable to provide raw data for individual pre-tRNAs rather than to lump all intron-containing tRNAs together.":

We assume that the reviewer refers to Figure 5d (in the current version changed to Figure 5c), as the comments pertain to the Hydro-tRNAseq analysis. In fact, each data point on the scatter plot refers to an individual intron-containing tRNA; that is, there is no lumping of groups of pre-tRNAs. We have reported normalized read counts for all intron-containing pre-tRNAs and their mature counterparts individually in Supplementary Table 8.

Response to "[T]here appears to be much scatter in the data (Extended data, Fig. 5a).":

That is correct and has been noted by us ([PMID: 28793268]; Cell Rep. 2017 Aug 8;20(6):1463-1475) and others [e.g. [PMID: 32796835]; Nat Commun. 2020 Aug 14;11(1):4104 and references therein). This probably reflects the variability in the expression dynamics and steady-state levels of individual pre-tRNAs. In addition, as we have shown previously, defects of pre-tRNA processing in patient-derived fibroblasts can be modest ([PMID: 24766809]; Cell. 2014 Apr 24;157(3):636-50.). This is in agreement with our current observations shown in Extended data, Fig. 5a and Figure 5d, where a modest increase in intron-containing pre-tRNAs is seen when comparing homozygotes to wild-type controls.

Response to "Extended data, Fig. 5b: Is there an explanation for the differences in splicing of pre-tRNAs encoded by different loci? This seems to be unexpected. Are the data statistically significant?":

Similar to the comment above, we have previously noted differences in the pre-tRNA levels among intron-containing tRNAs in cell lines ([PMID: 28793268]; Cell Rep. 2017 Aug 8;20(6):1463-1475.). Those results were observed in several replicates, and at much higher

sequencing depth than here. Therefore, we are confident that these differences are not artefactual, but represent the variability of pre-tRNA expression. In fact, a two-tailed paired Wilcoxon signed rank test comparing homozygotes and controls showed that the difference in the medians of the reported ratios of pre-tRNAs over mature tRNAs was statistically significant ($p < 0.0001$). We also repeated the statistical analysis excluding homozygote Ba1230, which showed the highest median of ratios, to exclude any potential outlier bias, and the difference in median ratios remained statistically significant ($p < 0.0001$). Thus, we conclude that the c.919G>T mutation results in modest, yet statistically significant increase in the ratio of pre-tRNA/mature tRNA for intron-containing tRNAs. We have added the necessary explanation in the text to reflect the response to the reviewer's comment:

'The distributions of the ratios of precursor over mature tRNA reads showed that there was no bias for the enrichment of any specific precursor tRNA among samples (Extended Data Fig. 5a and Supplementary Table 8). The difference in ratio medians between homozygous TSEN54 c.919G>T and control cell lines was statistically significant ($P < 0.0001$; two-tailed paired Wilcoxon signed rank test).'

5. Altered TSEN composition in cells with PCH mutations: Although the authors detected no differences in the in vitro assembly of the TSEN complex comparing WT enzyme to enzyme containing the PCH TSEN54 A307S mutation, assessment of this in vivo by pull-down of TSEN2 appears to result in reduced levels of co-purified PCH TSEN54 (Fig. 6b). These results are very interesting, but the data need to be quantitated for the ratio of IP vs input of TSEN54 and the IP for TSEN54 vs. nonspecific GAPDH and statistical significances should be reported. Nevertheless, the on-bead tRNA cleavage assay provides strong data in support of the hypothesis that TSEN complexes containing a TSEN54 PCH allele are defective in pre-tRNA cleavage, perhaps providing the strongest evidence that reduced pre-tRNA cleavage is causative for PCH; however, is it possible to normalize the data to levels of complete TSEN complexes?

The reviewer asked about quantification and statistical significance of the data provided in Fig. 6b. We now provide this information in Extended Data Fig. 6b. Values relative to TSEN subunit staining for controls (wild-type and heterozygous TSEN54 A307S) and homozygous TSEN54 A307S patients were pooled, and differences and statistical significance are reported. We could not, as we did for the northern blot analysis, repeat an IP panel identical to that shown in former Fig.6b, due to constraints in cultivating the required number of cells. To obtain enough non-immortalized fibroblasts from 14 cell lines to be able to perform the IP/assay described in Fig.6 required up to 6 months for cell pellet collection, which was not

possible to achieve in a time of restricted and regulated access to cell culture facilities. However, we would like to point out that for most cell lines, anti-TSEN34 immunoprecipitation followed by on-bead assay had been previously performed several times, with the same qualitative results as those presented in former Fig 6b. These experiments cannot, however, be used for statistical purposes, as they refer to separate trials aimed at optimizing the experimental set up, and differ in cell numbers, bead-antibody concentration, and IP conditions.

6. Discussion:

Cause of PCH: Since PCH cells appear not to have decreased levels of mature tRNAs, is it not possible that the phenotypes result from aberrant increased levels of pre-tRNAs? Perhaps this should be added to the other possibilities invoked. Discrimination of pre-tRNAs from mature tRNAs: The authors should elaborate upon what is meant by substrate selection resulting from different binding kinetics between pre-tRNAs and mature tRNAs. Perhaps, it would be valuable to mention that since splicing occurs in the nucleus of human cells, that the higher concentration of pre-tRNA substrate to mature tRNA in this organelle might contribute to specificity. The authors should also address the studies from Hayne et al. showing that tRNA fragments containing the anticodon stem loop + intron serve as substrates in vitro.

We have addressed the reviewer's notions in the results and discussion sections. Since we did not determine or compare on- or off-rates for pre-tRNAs and mature tRNAs to TSEN, we have excluded the statement about kinetic effects from the results section and focused the discussion as suggested by the reviewer as follows:

Results: *'Taken together, our results show that substrate recognition by human TSEN is primarily mediated by the mature domain of pre-tRNAs which positions intron-containing anticodon stems for cleavage.'*

Discussion: *'A higher concentration of intron-containing pre-tRNAs in the nucleus might contribute to TSEN substrate specificity. TSEN was shown to cleave artificial intron-containing anticodon stem loop structures³⁵. A three-dimensional structure of TSEN with substrate RNA will help define how eukaryotic TSEN recognizes pre-tRNAs and anticodon stem loop structures in particular^{35,42}.'*

Following the reviewer's remark, we now mention the potentially deleterious effects of pre-tRNA accumulation in neurons in the discussion.

'A secondary, potentially deleterious consequence of this failure could be the aberrant accumulation of pre-tRNAs, at levels much higher than those we measured in fibroblasts.'

Reviewer #2 (Remarks to the Author):

“Assembly defects of the human splicing endonuclease....”by Sekulovski et al

The Sekulovski paper focuses on the human 4-protein TSEN (tRNA splicing endonuclease) complex and the question of why mutations in the proteins of that complex are associated with pontocerebellar hypoplasia----a group of hereditary neuromuscular disorders. The TSEN complex associates with the RNA kinase CLP1, and roughly 6% of several hundred genome-encoded tRNAs need this system to create specific mature tRNAs.

Here the authors do a detailed biochemical/biophysical analysis of the recombinant WT and mutant complexes, together with an investigation of overall tRNA splicing events in patient fibroblasts. The work is of high quality and presented in a long-winded somewhat tedious fashion, as if the authors are trying to figure out how it all fits together. In the end, in spite of an enormous amount of quality work, they are not able to provide convincing clarity on why mutations in the proteins of TSEN complex are associated with pontocerebellar hypoplasia. And yet, much is learned about the biochemical personality and characteristics of the TSEN heterotetramer. With some further work with patient samples, which may inherently be difficult to achieve, this study could advance understanding closer to establishing a clear causal rationale for the disease phenotype.

We thank reviewer 2 for emphasizing the high quality of our work and what we have learned about the biochemical characteristics of the TSEN complex. We respond to the negatives in a comprehensive fashion below.

Positives

Confirm role in mammals of an A:I base pair for directing splice-site selection, as reported previously in *Drosophila*.

Do sophisticated protein manipulations to prepare and assemble a heterotetrameric TSEN2-15-34-54 stoichiometric complex and take that tetramer with CLP1 to give a heteropentameric complex. Impressive work.

Also, deconvoluted CLP1 and ATP from the rest of system. Did this by separately assembling an inactive heterodimeric TSEN2-54 and an inactive heterodimeric TSEN15-34 complex, which when combined together give endonuclease activity in the absence of CLP1 and ATP. Nice way to show heterotetrameric TSEN complex is necessary and sufficient for

cleavage activity, and that ATP and CLP1 have separate roles from catalysis per se on the endonuclease side of the overall splicing reaction.

Show that pre-tRNA substrate recognition is based on parts of pre-tRNA that are kept in the mature tRNA product, and that an intron per se is not needed for TSEN-pre-RNA complex stability.

Determined crystal structure of a stable proteolytic complex of the TSEN15-34 heterodimer. This required a laborious trial and error process to obtain a stable fragment. This dimer is missing substantial portions of the two constituent monomers of the heterodimer. They observe the well-established endonuclease fold. With further work they conclude that tRNA endonucleases for splicing have a common ancestor, which is a conclusion that contrasts with that of a prior solution NMR study of the homodimeric TSEN15 that evidently claimed just the contrary.

Construct 4 different, individual PCH-associated mutations in TSEN2, 34, and 54. In pull downs, the introduced mutations had no effect on the protein-protein interactions, subunit compositions, or precursor tRNA cleavage kinetics. They did however find that the mutant complexes had lower thermal stabilities, being diminished in their denaturation assays by 1 to 7 degrees, depending on the mutant complex.

They acquired patient fibroblasts harboring the most common PCH-associated mutation, namely, an A307S substitution in TSEN54. They establish that mutant cells have an increased level of precursor tRNAs, which of course are intron-containing. They also show that, in these cells, the TSEN assembly is defective when compared to wild-type patient fibroblasts. This defect is not due to a reduction in the levels of the respective proteins in the patient cells.

Negatives

While the work is carefully, thoughtful, and presented in well-written English, it reads as a collection of observations rather than as a cohesive story. I think the main disappointment is the lack of more convincing evidence that the PCH phenotype is caused by assembly defects in the TSEN complex, which in turn affect the output of mature tRNAs. What's missing are observations on patient fibroblasts beyond those harboring just the most common A307S substitution. Curiously, of the four recombinant mutant constructs they investigated, the A307S TSEN54 had the smallest effect on complex stability, as measured by thermal denaturation. Barely over 1 degree. And yet, the cells harboring this mutation had TSEN assembly defects. In contrast, the assembly of recombinant complexes was unaffected by this 'mild' or even by stronger mutations. One way to interpret this paradox is that, in the patients, the assembly defect and the negative effects on tRNA processing are due to factors

in cells that are yet uncharacterized and that possibly context effects are of over-riding importance. For example, the state of phosphorylation of TSEN? Or the disposition of CLP1? In the last paragraph of the Discussion the authors allude broadly to ambiguities. But they seem to overlook, or at least not comment on, the specific path forward that would help them here.

Also, the authors suggest that the severity of the disease may correlate with the magnitude of the lowered denaturation temperatures of the recombinant complexes of mutant proteins. I realize the difficulty, but if patient fibroblasts encoding the other mutant proteins were examined, then a correlation of the strength of the assembly defect in-step with the degree of pre-tRNA accumulation would provide nice support for the idea that, indeed, the lowered denaturation temperatures are close to, or at, the root cause of the overall disease phenotype.

I feel the reconstitution experiments, activity and melting curves analyses of the recombinant complexes, and the analysis of patient fibroblasts are the main story here. The x-ray structure of the TSEN15-34 complex is of less significance because so much of the complex is missing and only the truncated WT complex was examined. Also, the structure in some ways is confirmatory.

I agree that concluding the determined structure of the complex suggests it comes from a common ancestor, while contradicting an earlier conclusion by others, is useful and interesting. And yet really has no connection with the story here.

Although the Discussion is well written and covers a lot of ground, it does go on and on without a sharp cohesive focus.

I would like to see if the authors can get some additional patient fibroblasts and resubmit with a much 'tighter' story.

We are pleased that the reviewer appreciates our work and would like to point out that we do not claim to have clarified the mechanism of PCH development. Our study contains the first investigation of TSEN activity carried out in patient cells. While previous work relied on the over-expression of tagged constructs in heterologous cell lines, we established a system to unambiguously describe TSEN properties in PCH fibroblasts, complementing our in vitro biochemical and structural analysis of the mutant complex. We show by means of RNA analysis, protein immunoprecipitation, and pre-tRNA splicing assay that TSEN stability and activity is significantly compromised in these cells, although residual activity is sufficient to sustain tRNA production. We would like to stress the fact that these fibroblasts do not exhibit any obvious phenotype and have a growth rate comparable to that of controls, in line with the notion that the PCH phenotype is restricted to the degeneration of cerebellar neurons. Our results rule out the possibility that PCH mutations might impact TSEN activity exclusively in

the cerebellum, by inhibiting the binding of specific interactors or substrates. Instead, we demonstrate that these mutations interfere with complex assembly, stability, or enzymatic activity, leading to a general decrease of TSEN functionality that could be eventually exacerbated in specific cell/tissue contexts, such as cerebellar neurons. This is a novel perspective and, in our opinion, an advancement of knowledge in the field. Patient-derived neurons, obtained for example from fibroblasts by iPS technologies followed by reprogramming, will in the future be a better model to understand PCH development. However, these neurons would not be a suitable system for biochemical studies, as it would hardly be possible to obtain enough material to perform IP experiments, such as those we conducted in non-immortalized fibroblasts.

The crystal structure of the TSEN15/34 heterodimer allowed for the first time to rationalize the effect of one of the PCH mutations (TSEN15 H116Y) and lead us to correlate data on thermal stability with structure. As also highlighted by reviewers 1 and 3, the crystal structure is an important part of our work and, therefore, we would keep the logic structure of the manuscript as it is. We agree with the referee that it would be useful to include in the study also PCH patient-derived cells carrying mutations other than TSEN54 A307S, but, to our knowledge, such cells are unfortunately not available. Whether there is a precise, measurable correlation between the impact on thermal stability and the severity of the phenotype would most likely be difficult to prove. In the case of TSEN54 A307S, where we have a large cohort of patients and were able to identify a common trend, we registered considerable variation among cell lines, since they have different genomic backgrounds. In the case of other mutations, even if cells had been readily available, it might have been hard to obtain statistically significant data from 1-2 samples. However, as TSEN54 A307S is reported in over 90% of described PCH cases, we are confident that the defects in complex assembly and activity we describe for a cohort of several patients are indeed a hallmark of mutant TSEN.

We have re-shaped and streamlined the discussion section, as suggested by the reviewer, and have integrated our observations in the conceptual frame of the relevant literature. We discuss potential factors (not mutually exclusive) that could act as a neuron-specific trigger of PCH: 1) the need in neurons of a high tRNA/protein production rate; 2) the failure to splice one or more neuron-specific pre-tRNA(s); 3) the failure to bind cell-specific interactors/modifiers.

Reviewer #3 (Remarks to the Author):

To whom it might concern

The authors describe the recombinant production of tetrameric TSEN complex produced in human cells or in a baculovirus expression system. They biochemically analyze the interaction with substrate/product tRNAs. They map the substrate interaction to the mature domain of the target tRNA. The recombinant complex is active in intron excision. Neither 15/34 or 2/54 sub-dimers are active, which points towards a composite active site of the complex. The authors solve the crystal structure of the 15/34 dimer. A disease mutation lies in the interface of the dimer but does not disrupt dimer formation in vitro. However, this disease mutation, as well as other disease mutations tested, decrease the thermal stability of the complex. Furthermore, protein levels and pre/mature tRNA levels are compared in patient fibroblast samples. While overall protein levels remain undistinguishable, unspliced tRNA target species accumulate, which points towards intron excision defects in the patient samples. Co-IP points towards complex assembly defects in patient samples. The authors conclude that the thermal destabilization of the TSEN protein complex, paired with its low abundance leads RNA processing defects and cause some forms of pontocerebellar hypoplasia.

The study is sound, conclusive and technically comprehensive in itself. The protocol for recombinant production of the active TSEN complex is noteworthy and opens the way to further structural and mechanistical examination (p.ex. cryo-EM) of the mechanism of tRNA splicing and recognition of non-tRNA targets of the complex. In contrast to a previous study in *E. coli*, (Hayne et al. 2020), the authors report the existence of two stable sub-dimers in the eukaryotic expression systems. Biochemical testing of the substrate requirements has been performed by mutational analysis of the codon-anticodon interaction and comparison of pre- and mature tRNA. The main hypothesis, that thermal destabilization of the complexes leads to disease causing RNA processing defects, independent from CLP1, is convincingly cross validated by the analysis of patient samples.

We thank reviewer 3 for this very positive feedback.

Please address the following minor comments:

Line 55: “low copy number per cell” versus line 93: “High expression of TSEN54 in neurons of the pons, cerebellar dentate, and olivary nuclei” – this is contradictory, you have to relativize the context. Attention, in line 249 you cite different references than in 55.

We have changed the text to clarify this perceived contradiction. High expression was shown by histology by Budde et al. ([PMID: 18711368]; Nat Genet. 2008 Sep;40(9):1113-8.) when comparing different brain tissues, whereas Rauhut et al. ([PMID: 2211694]; J Biol Chem.

1990 Oct 25;265(30):18180-4.) calculated a total of ~100 molecules of TSEN per yeast cell from their purifications. We have deleted the sentence “High expression of TSEN54 in neurons of the pons, cerebellar dentate, and olivary nuclei” from the main text to avoid misunderstandings.

Line 57: challenged in which respect?

The NMR study by Song and Markley ([PMID: 17166513]; J Mol Biol. 2007 Feb 9;366(1):155-64.) has shown that human TSN15 forms homodimers at high protein concentration (0.5 mM) in solution and therefore suggested that TSEN might assemble in a different fashion as e.g. the archaeal homologues. We have modified the main text, which now reads as follows: ‘TSEN2–54 and TSEN15–34 are inferred to form distinct heterodimers from yeast-two-hybrid experiments, however, a solution NMR structure of homodimeric TSEN15 has challenged the proposed model of TSEN assembly^{8,9}.’

Line 193: Better: The domain swab is most likely a crystal-packing artefact, since the complex shows a homogenous 1:1 TSEN15:TSEN34 stoichiometric behavior in solution, as shown by SEC-MALS analysis.

The truncated TSEN15–34 heterodimer is monodisperse with a 1:1 stoichiometry after purification via SEC (Extended Data Fig. 3d). However, we could detect a tendency of the truncated TSEN15–34 heterodimer to further dimerize in solution at high protein concentrations, which we used for crystallization trials (as shown by SEC-MALS; Extended Data Fig. 3g). We therefore corrected the sentence to: ‘The domain swap is most likely a crystallization artifact, since TSEN15–34 migrates as a heterodimer during size exclusion chromatography (Extended Data Fig. 3d) and forms dimers of dimers at high protein concentration as shown by size exclusion chromatography multi-angle light scattering (SEC-MALS) (Extended Data Fig. 3g).’

Line 197: “C-terminal” instead of “N-terminal”

We have corrected the spelling mistake in line 197.

Figure 3 a, b, c, d : the figures are too small to see the side chain details. Moreover it is confusing how the different views relate to each other.

We have enlarged the panels and restructured Fig. 3 to show side chains in more detail. We have also indicated the rotations which relate the different views.

Some nice-to-have-discussed points, which I would not see as prerequisite for publication though:

Are there known human TSEN SNPs that are not malign, and would they keep the complex thermostable?

We share this interesting thought of reviewer 3. TSEN SNPs have not been investigated in more detail in the present study. Therefore, we cannot say whether they are destabilizing the complex or not.

Do all the known disease mutations lie in the complex interfaces?

Our crystal structure of the TSEN15-34 heterodimer revealed for the first time at high resolution a protein-protein interface, which is affected by a PCH disease mutation (TSEN15 H116Y). Despite structure predictions of short domains of TSEN subunits can be calculated based on homologies to archaeal endonucleases, no high-resolution structures of the entire complex are available to date. Therefore, it remains to be shown whether other disease mutations lie in complex interfaces. Since we see thermal destabilization of the TSEN complex by the PCH-related mutations in vitro, we speculate that the mutations affect directly or indirectly protein-protein interfaces of the complex.

How conserved are TSEN 15/34 to 2/54, would you imagine a symmetric dimer of dimers?

Despite the catalytic triad, which is very likely present in human TSEN2 and TSEN34, the subunits do not share much sequence conservation. Despite structural similarities of the C-terminal regions of the catalytic subunits, TSEN2 and TSEN34, and structural subunits, TSEN15 and TSEN54, true crystallographic symmetry can be excluded.

Could the domain-swab observed in the crystal structure resemble the mode of interaction between both TSEN heterodimers to form the tetramer? Extended Figure 1d would also indicate recombinant TSEN2/54 has the tendency to tetramerize.

We do not think that the domain swap is biologically relevant. It is conceivable that the short N-terminal α -helix/ β -hairpin element of the truncated TSEN34 protein, which we used for structure determination, is more flexible in isolation than in the tetrameric assembly of TSEN

and therefore tends to domain swap under crystallization conditions. A similar scenario could be envisioned for the isolated TSEN2/54 heterodimer, which –in our hands– has a tendency to multimerize at unphysiologically high protein concentrations in vitro. However, both structural and biochemical data suggest that tetramerization of eukaryotic TSEN is brought about by the L10 loop and the β -strand interactions in analogy to the archaeal endonucleases.

There are other studies linking decreased protein stability to disease that you could discuss/reference to support your observation.

Due to the suggestions of the other two reviewers and restrictions on the number of references, we have condensed and shortened the discussion. We apologize to the authors of the other studies for not being able to cite their interesting work here.

Please do not hesitate to contact me for further discussion.

Best regards,
Eva Kowalinski

Reviewers' Comments:

Reviewer #1:

Remarks to the Author:

In response to the referees' comments the authors have provided a thoughtful and rather thorough revised version of Sekulovski et al. The conclusions are important, well documented, and this publication will be of considerable interest to the RNA community.

There are only a few minor comments that should be addressed:

(1) Pg. 3, line 49 - there seems to be some contradiction regarding how many of the expressed tRNA^{Leu}CAA genes contain an intron; according to the publication cited by the authors, all expressed tRNAs are encoded by intron-containing genes, but according to the tRNA Santa Cruz database (<http://gtrnadb.ucsc.edu>), 5/6 tRNA^{Leu}CAA genes encode intron-containing pre-tRNAs. Perhaps the authors might make note of this, especially since they address possible different expression of tRNA genes in neuronal tissue.

(2) Pg. 5, line 115: Although the authors now address the related Hayne et al (2020) publication, the statement that Hayne et al. reported "reconstitution of TSEN/CLP1 from a bacterial expression host" is not quite correct as Hayne et al. also reconstituted the complexes from HEK mammalian cells. Their statement should be corrected.

Reviewer #2:

Remarks to the Author:

Re: "Assembly defects of the human tRNA splicing endonuclease contribute to impaired pre-tRNA processing in pontocerebellar hypoplasia" by Sekulovski, Devant, Panizza, ..., Martinez, Trowitzsch

Sekulovski, Devant, Panizza, et al present a mostly biochemical study on the human tRNA splicing endonuclease (TSEN) complex and how mutations can disrupt its function in pontocerebellar hypoplasia. They are able to reconstitute the active complex from purified subcomplexes and show its activity in the absence of CLP1. Structural elements on the tRNA were identified that determine splice site recognition. Impressively, they were able to obtain the crystal structure of one subcomplex. Finally, the authors investigate patient fibroblasts and find that enzymatic activity is not affected but instead that the mutations line the interaction interface between the enzymes and lower complex stability.

Overall, this is a thorough study with attention to detail. The reviews were already extensive and major issues have been cleared from the text but the following points need to be addressed:

- 1) It is up to the authors and editors but the title can be seen as misleading. While it is appreciated that the mutants were taken from patients and investigated due to their causative link with a disease state, they are only studied in the later part of the manuscript and only one mutant is tested in detail. Additionally, the data on the mutations are more speculative than the biochemical studies.
- 2) 4D: A307S barely differs in its Td from wildtype (50.4 vs 49.2C) but is the most commonly found disease causing mutant with strongly reduced pre-tRNA splicing in vitro. This is a clear disconnect in the data and either more convincing data on a destabilizing effect of the mutant should be provided or an alternative explanation entertained. The part in the discussion on alternative functions of TSEN is appreciated but it still follows the rationale of altered activity due to a less stable complex. Is for example the interaction with CLP1 affected instead?
- 3) If TSEN binds mature and pre-tRNA with comparable affinity and only exists in low numbers of 100 proteins per cell, how is pre-tRNA not outcompeted by mature tRNA, especially if there is no sequence specificity to binding and not all tRNAs are spliced? That would suggest strictly regulated and narrow

cellular localization.

4) Extended data 1C shows a band corresponding to the size of CLP1 in samples labeled to be TSEN only, same in 1B and E. Is it possible that *Drosophila* CLP1 was enriched during purification and found in the active complex in sub-stoichiometric amounts? A western blot should be able to detect small amounts of CLP1 or confirm its absence.

Reviewer #3:

Remarks to the Author:

My comments from the initial round of review (mainly on in vitro experiments) have been sufficiently addressed by the authors. I leave it to the two other reviewers to decide if their concerns regarding the analysis of the patient samples have been sufficiently addressed, since these lie out of my core expertise.

Detailed Responses to Reviewers' Comments

Reviewer #1 (Remarks to the Author):

In response to the referees' comments the authors have provided a thoughtful and rather thorough revised version of Sekulovski et al. The conclusions are important, well documented, and this publication will be of considerable interest to the RNA community.

We again thank reviewer #1 for the very positive evaluation of our work.

There are only a few minor comments that should be addressed:

(1) Pg. 3, line 49 - there seems to be some contradiction regarding how many of the expressed tRNA^{LeuCAA} genes contain an intron; according to the publication cited by the authors, all expressed tRNAs are encoded by intron-containing genes, but according to the tRNA Santa Cruz database (<http://gtrnadb.ucsc.edu>), 5/6 tRNA^{LeuCAA} genes encode intron-containing pre-tRNAs. Perhaps the authors might make note of this, especially since they address possible different expression of tRNA genes in neuronal tissue.

Thanks for this astute observation. The discrepancy in the number is based on the different methods used for detecting tRNAs. The Santa Cruz database is mostly derived from computational predictions of tRNA loci in the human genome. The cited publication (Gogakos T., et al., 2017, Cell Reports 20 (6), 1463-1475; PMID 28793268) provided experimental evidence by specialized RNA-sequencing protocols for a subset of the predicted tRNA loci. Specific attention had been paid towards the identification of tRNA splicing dependencies at that study. Therefore, at least in the experimental conditions of the cited work, which included deep sequencing of tRNA-enriched populations by two complimentary techniques, only intron-containing LeuCAA pre-tRNAs were detected (Fig. 5A in Gogakos T., et al., 2017).

(2) Pg. 5, line 115: Although the authors now address the related Hayne et al (2020) publication, the statement that Hayne et al. reported "reconstitution of TSEN/CLP1 from a bacterial expression host" is not quite correct as Hayne et al. also reconstituted the complexes from HEK mammalian cells. Their statement should be corrected.

We apologize for the incorrect statement; the text has been corrected and now reads: 'These data are in line with a recent study showing reconstitution of TSEN/CLP1 from bacterial and eukaryotic expression hosts ³⁵.'

Reviewer #2 (Remarks to the Author):

Re: "Assembly defects of the human tRNA splicing endonuclease contribute to impaired pre-tRNA processing in pontocerebellar hypoplasia" by Sekulovski, Devant, Panizza, ..., Martinez, Trowitzsch

Sekulovski, Devant, Panizza, et al present a mostly biochemical study on the human tRNA splicing endonuclease (TSEN) complex and how mutations can disrupt its function in pontocerebellar hypoplasia. They are able to reconstitute the active complex from purified subcomplexes and show its activity in the absence of CLP1. Structural elements on the tRNA were identified that determine splice site recognition. Impressively, they were able to obtain the crystal structure of one subcomplex. Finally, the authors investigate patient fibroblasts and find that enzymatic activity is not affected but instead that the mutations line the interaction interface between the enzymes and lower complex stability.

Overall, this is a thorough study with attention to detail. The reviews were already extensive and major issues have been cleared from the text but the following points need to be addressed:

We are pleased to read that all major issues raised by reviewer #2 were sufficiently addressed in the first revision of our manuscript.

1) It is up to the authors and editors but the title can be seen as misleading. While it is appreciated that the mutants were taken from patients and investigated due to their causative link with a disease state, they are only studied in the later part of the manuscript and only one mutant is tested in detail. Additionally, the data on the mutations are more speculative than the biochemical studies.

We intentionally structured the manuscript in such a way that the analyses of the PCH disease state in patient fibroblasts of the most prevalent mutation TSEN54 A307S are shown in the later part of the manuscript. The biochemical and structural data showcase a conundrum that sharp results on complex assembly and function using recombinant specimen in vitro do not fully explain the molecular events that may lead to a disease phenotype and that assays using patient material are necessary. We have long discussed the wording of the manuscript's title among the different groups involved and we still think that the title communicates the novel aspects of the paper in an appropriate manner. Therefore, we would suggest to keep the title of the manuscript as it is.

2) 4D: A307S barely differs in its Td from wildtype (50.4 vs 49.2C) but is the most commonly found disease causing mutant with strongly reduced pre-tRNA splicing in vitro. This is a clear disconnect in the data and either more convincing data on a destabilizing effect of the mutant should be provided or an alternative explanation entertained. The part in the discussion on alternative functions of TSEN is appreciated but it still follows the rationale of altered activity due to a less stable complex. Is for example the interaction with CLP1 affected instead?

We have shown that all recombinant TSEN complexes comprising PCH mutations show substantial reduced thermal stability in vitro. Furthermore, our atomic model of the TSEN15/34 interface provides structural evidence for the impact of a PCH disease mutation on complex stability. We specifically mention the small but significant difference in denaturation temperature of the TSEN54 A307S mutant complex in the manuscript.

Additionally, we find clear differences in complex assembly and activity in patient fibroblasts carrying the homozygous TSEN54 A307S mutation, when compared to healthy controls. In order to cope with the reviewer's concern, we now mention in the discussion that an additional effect of a PCH mutation might be a disturbed binding of mutant TSEN to CLP1 or may have – in addition to complex destabilization – effects on the location of mutant TSEN complexes within the cellular environment.

To cope with the reviewer's suggestion, we have added the following sentence to the discussion: '...Splicing of pre-tRNAs may require spatial regulation and local confinement. In this regard, altered complex stability might affect interactions between TSEN, CLP1, or other cellular components. ...'

3) If TSEN binds mature and pre-tRNA with comparable affinity and only exists in low numbers of 100 proteins per cell, how is pre-tRNA not outcompeted by mature tRNA, especially if there is no sequence specificity to binding and not all tRNAs are spliced? That would suggest strictly regulated and narrow cellular localization.

We fully agree with reviewer 2 that the data suggest that pre-tRNA splicing is strictly regulated with narrow cellular localization. Where splicing takes place in human cells is a very interesting question.

According to prior suggestions by reviewer #2 in the first revised version of the manuscript, we added the sentence 'A higher concentration of intron-containing pre-tRNAs in the nucleus might contribute to TSEN substrate specificity.' to the discussion and excluded the statement about potential kinetic effects for substrate recognition from the results section, since we did not determine on- or off-rates of pre-tRNAs or mature tRNAs to TSEN (we removed '...and suggests that discrimination between pre- and mature tRNAs might be dictated by kinetic effects.').

Again, to cope with the reviewer's suggestion, we have added the following sentence to the discussion: '...Splicing of pre-tRNAs may require spatial regulation and local confinement. In this regard, altered complex stability might affect interactions between TSEN, CLP1, or other cellular components. ...'

4) Extended data 1C shows a band corresponding to the size of CLP1 in samples labeled to be TSEN only, same in 1B and E. Is it possible that *Drosophila* CLP1 was enriched during purification and found in the active complex in sub-stoichiometric amounts? A western blot should be able to detect small amounts of CLP1 or confirm its absence.

*We assume that the reviewer is referring to *Spodoptera frugiperda* (S.f.) CLP1 and not *Drosophila* CLP1. To our knowledge, anti-S.f. CLP1 antibodies are not commercially available. We have tested an anti-human CLP1 antibody (Hanada et al., 2013, Nature 495, 474-480; PMID 23474986) in a cell lysate of Sf21 cells (wild-type) and a lysate of Sf21 cells infected with a maltose-binding protein (MBP)-CLP1-encoding baculovirus (Reviewer Figure 1a). Whereas the antibody reacts against the MBP-CLP1 chimera, no reactivity with other proteins of S.f. could be detected. More importantly, this*

reciprocal experiment showed that *S.f.* TSEN did not copurify with human MBP-CLP1 to an extent detectable by Coomassie blue stain (Reviewer Figure 1b). We would also like to point out that the protein, which corresponds to the faint band visualized in our TSEN preparations without CLP1 (Extended Data Figure 1), also migrates slightly differently in SDS-PAGE as compared to human CLP1. We therefore argue that this band rather corresponds to a contamination than to *S.f.* CLP1. Eventually, even if the band corresponded to *S.f.* CLP1 it would not change the results as presented in Extended Data Figure 1.

Reviewer Figure 1 | Purification of a recombinant maltose-binding protein (MBP)-CLP1 fusion construct produced in *Spodoptera frugiperda*. Purification steps of an MBP-CLP1 fusion construct via Amylose resin were visualized by immunoblotting using an anti-human CLP1 antibody (**a**) from control Sf21 cells or Sf21 cells infected with an MBP-CLP1 encoding baculovirus, and by SDS-PAGE with subsequent Coomassie blue stain (**b**). The recombinant His₁₀-MBP-CLP1 fusion construct is indicated on the right. M – protein size marker, L – total cell lysate, SN – cleared cell lysate after centrifugation, E – eluate.

Reviewer #3 (Remarks to the Author):

My comments from the initial round of review (mainly on in vitro experiments) have been sufficiently addressed by the authors. I leave it to the two other reviewers to decide if their concerns regarding the analysis of the patient samples have been sufficiently addressed, since these lie out of my core expertise.

We again thank reviewer #3 for her very positive evaluation of our work.